# VIDEO-GPT VIA NEXT CLIP DIFFUSION

**Shaobin Zhuang**[1,*]   **Zhipeng Huang**[2,*]   **Ying Zhang**[4]   **Fangyikang Wang**[3,*]
**Canmiao Fu**[4]   **Binxin Yang**[4]   **Chong Sun**[4]   **Chen Li**[4]   **Yali Wang**[5,6,†]

[1]Shanghai Jiao Tong University   [2]University of Science and Technology of China
[3]Zhejiang University   [4]WeChat Vision, Tencent Inc.
[5]Shenzhen Key Lab of Computer Vision and Pattern Recognition, Shenzhen Institutes of
Advanced Technology, Chinese Academy of Sciences   [6]Shanghai AI Laboratory

{hahahahaha}@sjtu.edu.cn   {ethanphuang, yinggzhang}@tencent.com
{kenelmfu, binxinyang, waynecsun, chaselli}@tencent.com
wangfangyikang@zju.edu.cn   yl.wang@siat.ac.cn

## ABSTRACT

GPT has shown its remarkable success in natural language processing. However, the language sequence is not sufficient to describe spatial-temporal details in the visual world. Alternatively, the video sequence is good at capturing such details. Motivated by this fact, we propose a concise Video-GPT in this paper by treating video as new language for visual world modeling. By analogy to next token prediction in GPT, we introduce a novel next clip diffusion paradigm for pretraining Video-GPT. Different from the previous works, this distinct paradigm allows Video-GPT to tackle both short-term generation and long-term prediction, by autoregressively denoising the noisy clip according to the clean clips in the history. Extensive experiments show our Video-GPT achieves the state-of-the-art performance on video prediction, which is the key factor towards world modeling (Physics-IQ Benchmark: **Video-GPT 34.97** *vs.* **Kling 23.64** *vs.* **Wan 20.89**). Moreover, it can be well adapted on 6 mainstream video tasks in both video generation and understanding, showing its great generalization capacity in downstream.

## 1 INTRODUCTION

Over the past few years, natural language processing has been mainly driven by training large language models (LLMs) from web-scale text data. In particular, GPT series Radford & Narasimhan (2018); Radford et al. (2019); Brown et al. (2020); Achiam et al. (2023) have demonstrated remarkable generalization capabilities on various downstream language tasks, which further opens the possibility towards general artificial intelligence. However, the language sequence is often good at expressing high-level abstractions, while it may not be sufficient to capture rich spatial-temporal details in the visual world Liu et al. (2024); Yang et al. (2024b). For example, it is difficult to describe how to tie a knot by language. Alternatively, the video sequence has been considered as a preferred candidate to describe such details, since it can record visual knowledge of our dynamical world at different spatial and temporal resolutions Bai et al. (2023); Liu et al. (2024). Hence, there is a natural question: *Can we treat video as new language for visual world modeling?*

The recent studies have shown that, video generation is a promising direction to achieve this goal OpenAI (2024); Bruce et al. (2024); Kondratyuk et al. (2023). The typical design is video diffusion by adding noise and denoising gradually Song et al. (2021); Ho et al. (2020). Although such a paradigm has achieved significant progress Bar-Tal et al. (2024); Kong et al. (2024); Kuaishou (2024); Wang et al. (2025), it often suffers from difficulty in long-term future prediction that is a critical factor of world modeling Ouyang et al. (2024); Li et al. (2024). To address this problem, the autoregressive attempts have been made for long-context video modeling, on analogy of next token prediction in LLMs Kondratyuk et al. (2023); Liu et al. (2024); Bai et al. (2023). However,

---

*Work done as interns at WeChat Vision, Tencent Inc.
†Corresponding author.

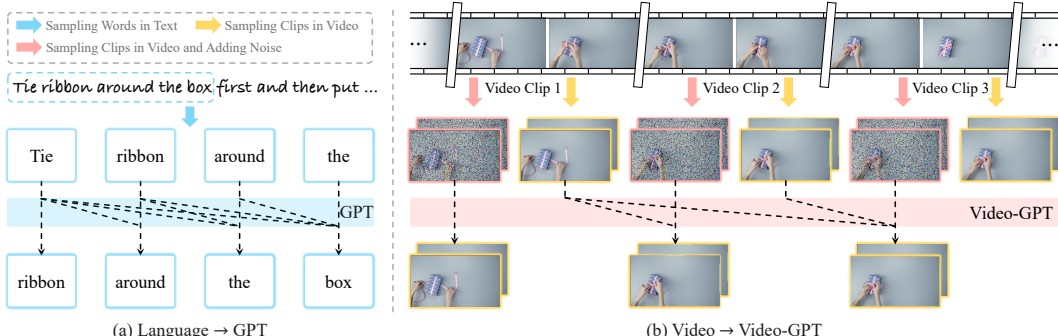

Figure 1: **Next clip diffusion.** We draw an analogy with GPT's next token prediction and model each video clip as a visual word by denoising the next noisy clip, conditioning on the previous video.

compared with its diffusion counterpart, the generation performance of this paradigm still needs to be further improved. In order to leverage advantages from both paradigms, several approaches combine diffusion and autoregressive modeling in a unified transformer style Xiao et al. (2024); Deng et al. (2024); Zhou et al. (2024); Huang et al. (2025b); Hu et al. (2024); Deng et al. (2025). But these design mainly works on the image domain, without insightful analogy between language and video.

To fill this gap, we propose a concise Video-GPT, by analogy with generative pretraining design in GPT. Inspired by next token prediction in GPT, we introduce a novel *next clip diffusion* paradigm for our Video-GPT. Specifically, we creatively treat a *clip* in the video as the role of a *word* in the language, since both of them describe local temporal information, respectively, in the video and language sequence. However, different from a discrete word token, it is often challenging to predict a continuous video clip in the next step. To address this difficulty, we introduce a flexible diffusion design within an interleaved sequence of noisy and clean clips in the temporal order. As shown in Fig. 1, one can randomly sample a number of clips from a training video. For each clip, we generate its noisy clip by adding noise on it, based on the forward process of diffusion. Then, we re-arrange noisy and clean clips in temporal order, constructing an interleaved clip sequence. Consequently, for each noisy clip, we leverage the clean clips in the previous as context, and reconstruct the corresponding clean clip by the diffusion loss. In this case, our Video-GPT inherits advantages from both GPT and diffusion, but within the basic unit of a video clip. This allows it to capture long-term video prediction as well as short-term video generation for effective generative pretraining on videos.

Our contributions can be summarized as follows. *First*, we introduce a concise and self-supervised pretrained Video-GPT, by analogy to GPT. Different from GPT, our Video-GPT aims at modeling rich spatial-temporal information in the visual world. *Second*, we design a novel next clip diffusion paradigm for pretraining Video-GPT. Different from the previous works mentioned above, this hybrid paradigm allows Video-GPT to tackle both short-term generation and long-term prediction, via integrating diffusion and autoregressive design in the video clip level. Finally, our pretrained Video-GPT achieves state-of-the-art performance on Physics-IQ Benchmark and Kinetics-600, fully demonstrating its potential for visual world modeling. After fine-tuning on 6 downstream video generation and understanding tasks, it also achieved preferable performance for generalization.

## 2 RELATED WORK

**Video Diffusion Models.** With diffusion models demonstrating remarkable performance in image generation Saharia et al. (2022); Rombach et al. (2021); Dhariwal & Nichol (2021); Ramesh et al. (2022), video diffusion models Ho et al. (2022) also surpasses traditional methods Clark et al. (2019); Saito et al. (2018); Kahembwe & Ramamoorthy (2019) in video generation. With the emergence of large-scale text–video datasets Chen et al. (2024b); Wang et al. (2024c; 2023c), text-to-video diffusion models Blattmann et al. (2023); Zhuang et al. (2024); Yang et al. (2024c); OpenAI (2024); Wang et al. (2025); Kong et al. (2024); Wang et al. (2023b) exhibit realistic generation capabilities and strong transferability to downstream tasks Yue et al. (2025); Hu & Xu (2023). However, the supervised paradigm based on text annotations faces two challenges as it scales up. **1) Quantity**

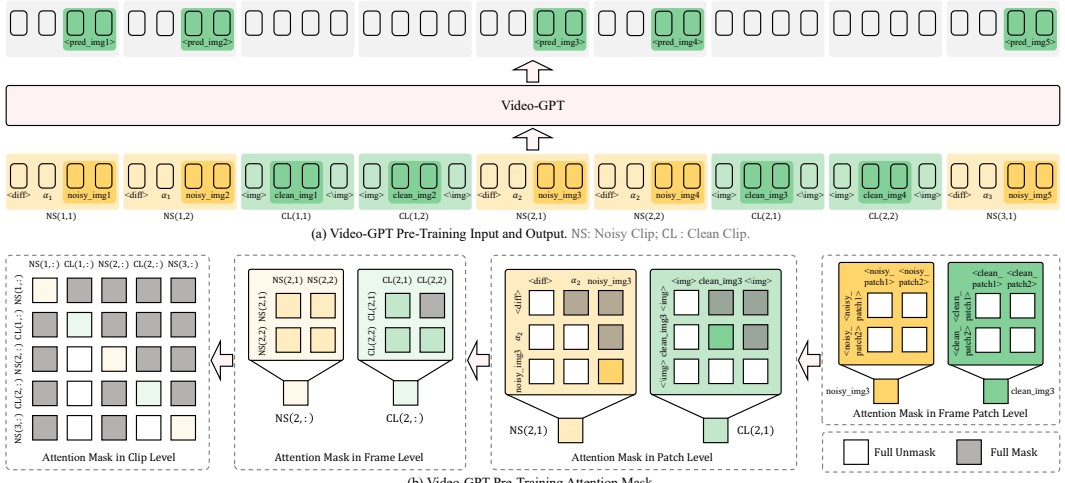

Figure 2: **Video-GPT pretraining framework.** The full attention mask is shown in Fig. 10

Table 1: **Progressive training strategy.**

| Stage | Resolution | Frame Num. | Frame Interval | Clip Num. | Train Steps |
|---|---|---|---|---|---|
| 1 | Flexible | 16 | 4 | 16 | 300000 |
| 2 | Flexible | 48 | 4 | $\sim \text{Uniform}\{2, 3, \cdots, 48\}$ | 25000 |
| 3 | Flexible | 48 | $\sim \text{Uniform}\{4, 5, \cdots, 12\}$ | $\sim \text{Uniform}\{2, 3, \cdots, 48\}$ | 40000 |
| 4 | Flexible | 80 | $\sim \text{Uniform}\{4, 5, \cdots, 12\}$ | $\sim \text{Uniform}\{2, 3, \cdots, 80\}$ | 20000 |

**of Annotated Data**: Current methods require meticulous data filtering and annotation processes Menapace et al. (2024); Polyak et al. (2024), resulting substantially smaller data scale compared to LLM Dubey et al. (2024); Yang et al. (2024a); Kaplan et al. (2020). **2) Quality of Annotated Data**: Text cannot fully capture the meaning of a video. While increasing the granularity of descriptions Chen et al. (2023a); Betker et al.; Segalis et al. (2023) alleviates this issue, the difficulty of annotating such descriptions significantly increases as video length scales up.

**Autoregressive Video Models.** ImageTransformer Parmar et al. (2018) migrated next token prediction to image generation in pixel space. Chameleon Team (2024) and LlamaGen Sun et al. (2024) perform pretraining on images in latent space. LVM Bai et al. (2023) and VideoWorld Ren et al. (2025) reveal that models can acquire world knowledge from videos. LWM Liu et al. (2024) compresses videos into latent space for pretraining, but focuses on video understanding capabilities. In general, next token prediction based visual generation Wang et al. (2024d) still underperform compared with the most advanced diffusion models Labs (2024); Liao et al. (2025); Wang et al. (2025); Kong et al. (2024). In contrast, our Video-GPT adopts a next clip diffusion paradigm, which diffusion generation within video clips and applies autoregressive generation between clips. It enhances video generative modeling while utilizing historical information as a self-supervised condition.

**Video Generation Models with Hybrid Modeling.** Recently, video generation models that hybridize autoregressive and diffusion modeling have begun to emerge. Self-forcing Huang et al. (2025a) is based on the paradigm of next frame prediction, using diffusion for generation within each frame, and employing autoregression between frames. APT2 Lin et al. (2025) introduces the autoregressive adversarial post-training to achieve real-time video generation. Video-GPT differs from them in two fundamental ways: 1) **Self-Supervised Pre-Training:** Video-GPT is pre-trained through self-supervised pre-training similar to LLM, does not require any annotations other than videos in this process. Through pre-training on the open source Panda-70M dataset Chen et al. (2024b), we achieve and surpass the physical world modeling capabilities of the top closed-source models like Gen3 Runway (2024) and Sora OpenAI (2024). 2) **Next Clip Diffusion:** Video-GPT is autoregressive at the clip level. This is more efficient and flexible as it allows parallel, bidirectional processing (diffusion) within a multi-frame clip, but maintains temporal order between clips.

## 3 VIDEO-GPT

In this section, we first introduce our Video-GPT via next clip diffusion. Subsequently, we describe how to adapt it for typically downstream tasks in both video generation and understanding. For better understanding, we also present the pseudocode of Video-GPT in Sup. A.

### 3.1 INPUT: CLIP SEQUENCE CONSTRUCTION

As mentioned before, we notice that a clip in the video plays a similar role as a word in the language. Hence, we first divide a training video into a number of clips to construct the basic processing unit in our Video-GPT. Specifically, we uniformly sample $N$ frames from each training video. Then, we randomly divide these frames into $K$ clips, where $K$ is sampled by $K \sim \text{Uniform}\{2, 3, \cdots, N\}$.

**Forward Diffusion Process on Each Clip**. Since diffusion has shown the powerful capacity of video generation, we propose to leverage it to predict the next video clip based on the previous videos. According to this reason, we perform the forward diffusion process on latent of each clip by adding Gaussian noise. Specifically, for the $k$-th clip, we first apply continues VAE Rombach et al. (2021) to compress each frame in this clip, and patchify it to obtain the latent feature of each frame (i.e., the feature matrix of all the patch tokens in this frame). Next, we choose flow matching Lipman et al. (2022) for diffusion, due to its efficiency. According to flow matching, we add the noise on the latent feature in the following way,

$$\Psi(k, i, \alpha_k) = \alpha_k \Phi(k, i) + (1 - \alpha_k)\varepsilon_{k,i}, \tag{1}$$

where $\Phi(k, i)$ is the latent feature of the $i$-th frame in the $k$-th clip, and $\Psi(k, i)$ is the noisy feature of this frame. The weight $\alpha_k$ is sampled by $\alpha_k \sim \text{Uniform}[0, 1]$, and the noise $\varepsilon_{k,i}$ is sampled by $\varepsilon_{k,i} \sim \mathcal{N}(\mathbf{0}, \mathbf{I})$. Note that, in order to inference the content in the video clip in parallel, we apply the same $\alpha_k$ for all frames in the video clip during training.

**Noise-Clean Interleaved Clip Sequence**. After getting the noisy clips, we next arrange the input sequence for our Video-GPT. By analogy of GPT, we leverage the historical clips as temporal context to denoise the noisy clip in the next step. But instead of using the noisy clips in the history Chen et al. (2024a); Yi et al. (2025), we leverage the original clean clips in the history, in order to denoise the next clip conditioned on the clean and correct context. Hence, we feed both noisy and clean clips as input to our Video-GPT. To distinguish different frames in these clips, we add extra tokens for specializing them. **1) Clip-Level** Token Form of Clean Clip: For each clean clip, we add the boundary tokens with the latent feature of each frame in this clip,

$$\mathbf{CL}(k, i) = [\texttt{}, \Phi(k, i), \texttt{<\img>}]. \tag{2}$$

Hence, the clip-level token form of the $k$-th clean clip is $\mathbf{CL}(k, :) = [\mathbf{CL}(k, 1), ..., \mathbf{CL}(k, N_k)]$, where $N_k$ is the number of frames in this clip. **2) Clip-Level** Token Form of Noisy Clip: For each noisy clip, we add two extra tokens with the noisy latent feature of each frame in this clip,

$$\mathbf{NS}(k, i) = [\texttt{<diff>}, \alpha_k, \Psi(k, i, \alpha_k)]. \tag{3}$$

where <diff> is a denoising hint token to indicate that $\Psi(k, i)$ is noisy. Moreover, we also add the weight $\alpha_k$ for the $k$-th noisy clip, in order to provide the timestep information in the flow matching. As a result, the clip-level token form of the $k$-th noisy clip is $\mathbf{NS}(k, :) = [\mathbf{NS}(k, 1), ..., \mathbf{NS}(k, N_k)]$. **3) Noise-Clean Interleaved Input**: As shown in Fig. 2 (a), after formulating both clean and noisy clips, we arrange them together as the input sequence to Video-GPT. Since we aim at sequential modeling, we take a pair of noisy and clean clips as a group, and arrange these groups in the temporal order. As a result, we obtain an interleaved sequence of noisy and clean clips as input, i.e.,

$$\mathbf{Input} = [\mathbf{NS}(1, :), \mathbf{CL}(1, :), ...., \mathbf{NS}(k, :), \mathbf{CL}(k, :), ..., \mathbf{NS}(K, :)]. \tag{4}$$

### 3.2 PRETRAINING: NEXT CLIP DIFFUSION

After obtaining the input sequence, we feed it into our Video-GPT for generative pretraining. By analogy to GPT, we use the vanilla transformer architecture for conciseness. The next question is how to build up relations between noisy and clean clips for next clip diffusion. As shown in Fig. 2 (b), we introduce a hierarchical masking method to indicate such relations in the attention operation.

Figure 3: **Video-GPT inference framework.** We iteratively denoise the 2nd noisy clip $\mathbf{NS}(2,:)$ to $\mathbf{CL}(2,:)$, and use it along with the 1st clean clip $\mathbf{CL}(1,:)$ to condition the prediction of the 3rd noisy clip $\mathbf{NS}(3,:)$. The number of frames in each clip can also vary during inference.

**Clip-Level Mask**. As our input is a clip sequence, we first need to define the dependence among clips. By analogy to the dependence of words in GPT, we basically apply a causal relation as clip dependence. **1) Clean Clip Mask**. As shown in Fig. 2 (b), the $k$-th clean clip depends on itself and the previous $(k-1)$ clean clips. **2) Noisy Clip Mask**. Alternatively, the $k$-th noisy clip depends on itself for sure. However, instead of depending on the previous $(k-1)$ noisy clips, it depends on the previous $(k-1)$ clean clips. The reason is that, these clean clips in the history provide the correct temporal context for denoising the $k$-th noisy clip. It is the core idea in our next clip diffusion.

**Frame-Level Mask**. Since each clip consists of several frames, we need to further specify the dependence among frames in the defined clip-level mask. **1) Clean Frame Mask**. As shown in Fig. 2 (b), for the $i$-th frame in the $k$-th clean clip, it depends on itself, the $(i-1)$ frames in this clean clip, and all the frames in the previous $(k-1)$ clean clips. **2) Noisy Frame Mask**. Alternatively, for the $i$-th frame in the $k$-th noisy clip, it depends on itself, all the other frames in this noisy clip, and all the frames in the previous $(k-1)$ clean clips. Note that, different from the causal design for the clean frame, the $i$-th noisy frame depends on all the other frames in the same noisy clip. The reason is that iterative denoising turns noisy clips into clean clip as history condition, the mask of noisy frames will not affect later inference and bidirectional attention can better enhance generation quality Yin et al. (2024); Kim et al. (2021).

**Patch-Level Mask**. As shown in Eq. (2) and (3), each frame (clean or noisy) is formulated by extra hint tokens, and the latent feature of this frame which actually refers to the feature matrix of all patch tokens in this frame. Hence, we need to further specify the dependence among these hint tokens and patch tokens in the defined frame-level mask. **1) Clean Patch Mask**. As shown in Fig. 2 (b), we follow the similar design of GPT, the dependence is causal among tokens, i.e., , $\Phi(k,i)$, and <\img>. But differently, the clean frame feature $\Phi(k,i)$ is actually the feature matrix of all the patch tokens in this frame. Since the patch tokens describe spatial relation rather than temporal one, we define the full dependence among these patch tokens. **2) Noisy Patch Mask**. Similar to clean patch mask, we define the causal dependence among tokens, i.e., <diff>, $\alpha_k$, and $\Psi(k,i)$. We also define the full dependence among patch tokens in noisy frame as the same as the one in clean frame.

**Training Target.** Based on this hierarchical masking, our Video-GPT can effectively leverage the masked attention to denoise the $k$-th noisy clip in the next step, according to the previous $(k-1)$ clean clips of input sequence. Subsequently, we compute the $\mathcal{L}_2$ loss between each denoised clip feature and its corresponding clean clip feature to pretrain our Video-GPT. It is worth mentioning that, different from previous diffusion works, we employ Video-GPT to predict the video clip directly instead of noise Chen et al. (2023a); Wang et al. (2023b) or velocity Esser et al. (2024); Yang et al. (2024c), to keep the training setting as simple as possible, allowing to easily adapt the pretrained Video-GPT for various downstream tasks.

**Progressive Training.** As the length of the video frame increases, the computational cost of the attention operation will increase quadratically. To alleviate such computation, we leverage a simple but effective progressive training strategy. Specifically, we start training from short videos to long videos. As shown in Tab. 1, the reason is that in the initial pretraining stage we use 16 frames—with each clip containing only one frame for next-frame prediction—then gradually increase both the number of clips and frames per clip for Video-GPT pretraining.

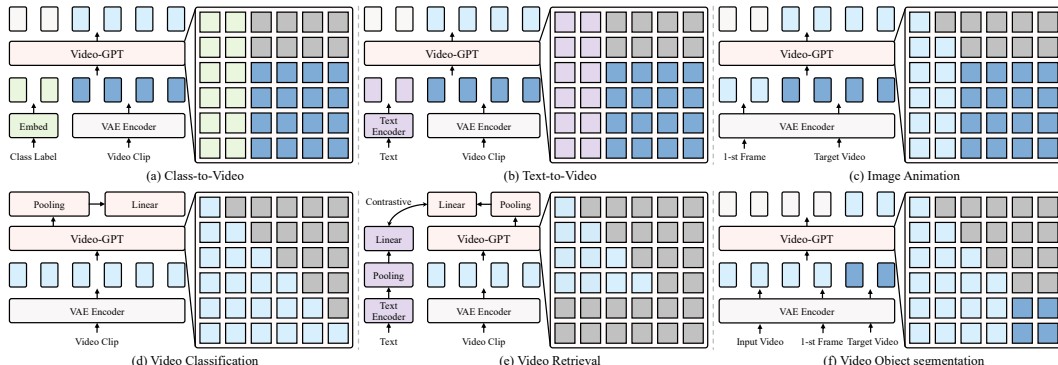

Figure 4: **Fine-tuning Video-GPT on downstream tasks.**

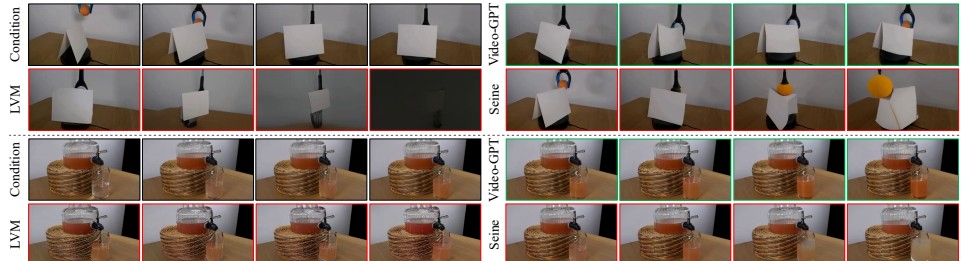

Figure 5: **Qualitative results on Physics-IQ Benchmark.** The videos predicted by our Video-GPT based on condition frames are more consistent with physical laws than other methods.

## 3.3 Inference: Autoregressive Video Prediction

After pretraining Video-GPT, one can naturally leverage it for video prediction, by analogy to GPT. As shown in Fig. 3 (a), video prediction can be autoregressively performed in the inference phase. Specifically, we treat the previous $k$ denoised clips (outputs of Video-GPT in the history) as the clean clips, and leverage them as temporal context to denoise the $(k + 1)$ noisy clip,

$$\mathbf{DNS}(k + 1, :) = \mathbf{Video\text{-}GPT}\Big( \mathbf{DNS}(1, :), ..., \mathbf{DNS}(k, :), \ \mathbf{NS}(k + 1, :) \Big), \qquad (5)$$

where $\mathbf{DNS}(k + 1, :)$ is the $(k + 1)$-th denoised clip. Additionally, $\mathbf{DNS}(1, :)$ is a clean clip which is initially given as input for video prediction. Note that, the input of Video-GPT in this inference phase is a causal formulation. Hence, the clip-level mask is defined as the causal one for each iteration of autoregressive prediction, as shown in Fig. 3 (b). We use all previously generated clips as context, up to the maximum sequence length supported by model pre-training. To generate longer videos than fit within this context window, a standard sliding window approach is adopted.

## 3.4 Generalization on Downstream Video Tasks

As we noticed, GPT has shown its powerful generalization capacity on various downstream NLP tasks. Hence, we expect that Video-GPT can show the similar capacity on various downstream video tasks. For this reason, we adapt Video-GPT to address typical downstream video tasks in Fig. 4, covering both video generation and understanding. For video generation, we focus on class-to-video generation, text-to-video generation, and image animation. For video understanding, we focus on video classification, video retrieval, and video object segmentation.

**Class-to-Video Generation and Text-to-Video Generation.** As shown in Fig. 4, we use a similar processing mode for these two types of tasks, since both generation tasks of Video-GPT are conditioned on extra tokens (either from a category tag or a video description). Hence, we introduce a causal attention between extra tokens and clip tokens, i.e., a extra token only depends on other extra tokens, while a clip token depends on both extra and clip tokens.

**Image Animation and Video Object Segmentation**. Since both tasks are to predict video content conditioned on the given sequence, we discuss them together here. For image animation, we are given the first frame. The goal is to predict the rest video conditioned on the first frame. For video

Table 2: **Quantitative comparison of models evaluated on the Physics-IQ Benchmark.**

| Model | w/o. Text | Spatial IoU↑ | Spatio Temporal↑ | Weighted Spatial IoU↑ | MSE↓ | Phys. IQ Score↑ |
|---|---|---|---|---|---|---|
| Sora (I2V) OpenAI (2024) | ✗ | 0.138 | 0.047 | 0.063 | 0.030 | 10.00 |
| Pika 1.0 (I2V) PikaLabs (2024) | ✗ | 0.140 | 0.041 | 0.078 | 0.014 | 13.00 |
| SVD (I2V) Blattmann et al. (2023) | ✓ | 0.132 | 0.076 | 0.073 | 0.021 | 14.80 |
| LVM (V2V) Bai et al. (2023) | ✓ | 0.100 | 0.147 | 0.077 | 0.021 | 18.02 |
| Lumiere (I2V) Bar-Tal et al. (2024) | ✗ | 0.113 | 0.173 | 0.061 | 0.016 | 19.00 |
| Open-Sora-Plan v1.3.0 (I2V) Lin et al. (2024) | ✗ | 0.142 | 0.139 | 0.074 | 0.021 | 19.42 |
| Wan2.1 (I2V) Wang et al. (2025) | ✗ | 0.153 | 0.100 | 0.112 | 0.023 | 20.89 |
| HunyuanVideo Kong et al. (2024) | ✗ | - | - | - | - | 22.36 |
| Gen 3 (I2V) Runway (2024) | ✗ | 0.201 | 0.115 | 0.116 | 0.015 | 22.80 |
| Lumiere (V2V) Bar-Tal et al. (2024) | ✗ | 0.170 | 0.155 | 0.093 | 0.013 | 23.00 |
| Kling1.6 (I2V) Kuaishou (2024) | ✗ | 0.197 | 0.086 | 0.144 | 0.025 | 23.64 |
| LTX-Video-I2V HaCohen et al. (2024) | ✗ | - | - | - | - | 26.80 |
| Cogvideo-I2V-5B Hong et al. (2022) | ✗ | - | - | - | - | 27.1 |
| Seine (V2V) Chen et al. (2023b) | ✗ | 0.163 | 0.208 | 0.131 | 0.012 | 29.13 |
| VideoPoet (V2V) Kondratyuk et al. (2023) | ✗ | **0.204** | 0.164 | 0.137 | 0.010 | 29.50 |
| **Video-GPT (V2V)** | ✓ | 0.198 | **0.240** | **0.144** | **0.007** | **34.97** |

Table 3: **Quantitative comparison of video generation models evaluated on the Kinetics-600.**

| Model | Parameters | Architecture | Text Encoder | FVD(500)↓ | FVD(5000)↓ |
|---|---|---|---|---|---|
| Open-Sora-Plan v1.3.0 Lin et al. (2024) | 2B | DiT | mT5-XXL Xue et al. (2020) | 384.27 | 95.36 |
| LVM Bai et al. (2023) | 7B | Vanilla Transformer | - | 356.48 | 94.61 |
| Seine Chen et al. (2023b) | 0.9B | U-Net | CLIP-ViT-L Radford et al. (2021) | 332.80 | 91.08 |
| **Video-GPT** | 3.8B | Vanilla Transformer | - | **315.40** | **89.44** |

Table 4: **Inference setting ablation.**

| | Number of Frames in Each Video Clip | | | | | | | History Conditioned Classifier-Free Guidance | | | | | |
|---|---|---|---|---|---|---|---|---|---|---|---|---|---|
| Hyperparams. | 1 | 2 | 4 | 8 | 16 | **32** | 54 | 1 | 1.5 | 2.0 | 2.5 | **3.0** | 3.5 |
| Phy. IQ Score | 0.00 | 6.21 | 18.44 | 29.90 | 32.86 | **34.94** | 32.18 | 25.09 | 31.96 | 32.22 | 32.65 | **32.72** | 31.88 |

Table 5: **Training setting ablation.**

| | Pretraining Paradigm | | Pretraining Frame Number | | | Add Noise to Clean Clip | | Dataset Scale | |
|---|---|---|---|---|---|---|---|---|---|
| Hyperparams. | Next Token Prediction | Next Clip Diffusion | 16 | 48 | **80** | NO | **YES** | 1M | **70M** |
| Phy. IQ Score | 21.59 | **34.94** | 22.06 | 33.09 | **34.94** | 32.54 | **33.09** | 23.16 | **33.09** |

object segmentation, we are given a video and the first frame with the masked objects. The goal is to track the masked objects throughout the video. To address these tasks by fine-tuning, we leverage the input frame or video as the clean clip in our Video-GPT, and the video to be generated as the noisy clip. The mask strategy follows the settings in Sec. 3.2.

**Video Classification and Video Retrieval.** For video classification, we pool clean clip features from Video-GPT for linear probing. For video retrieval, we use the clean clip feature for contrastive learning with the text feature from a text encoder, e.g., CLIP-ViT-L-Patch14 Radford et al. (2021).

# 4 EXPERIMENTS

## 4.1 IMPLEMENTATION

**Dataset.** In pretraining, we employ Panda-70M Chen et al. (2024b) as the training dataset. In order to vertify the impact of data scale on model performance, we also employ OpenVid-1M Nan et al. (2024) for ablation. For downstream tasks, we fine-tune the model with supervised datasets. Specifically, for class-to-video and video classification, we conduct experiments on UCF-101 Soomro et al. (2012). For text-to-video generation, we fine-tune the model with OpenVid-1M, and for the video retrieval task, we used videos and annotated texts from the Panda-70M. For video object segmentation, we randomly select 1000 cases from GetIn-1M Zhuang et al. (2025). Notably, for image animation—which is in high demand within the industry, there is no commonly recognized benchmark. Therefore, we collect three datasets from the internet to evaluate the generalization capability of pretrained Video-GPT on image animation. These datasets correspond to the categories of *lie down*, *transform*, and *fly*, with each dataset containing fewer than one hundred videos.

**Settings.** In this paper, we use VAE from SDXL Podell et al. (2023), with Video-GPT inheriting the model architecture from Phi-3-mini Abdin et al. (2024). We additionally incorporated

Figure 6: **Qualitative results of Video-GPT on class-to-video generation on UCF-101.**

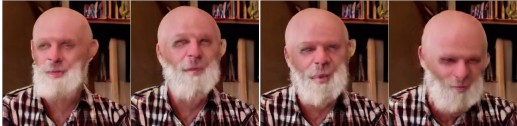

"The video features a man with a white beard and a bald head, wearing a plaid shirt. He is seated in a room with a bookshelf in the background. The man appears to be in deep thought or contemplation, as he gazes off to the side. The room has a warm and cozy atmosphere, with the bookshelf filled with various books and papers."

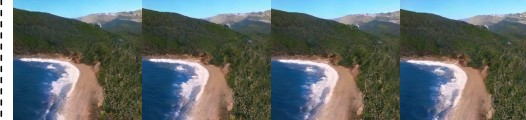

"A aerial view of a beach nestled in a cove. The beach, with its light brown sand, meets the deep blue water at the edge. On the right side of the cove, there's a hill. Above all this, the sky is a bright blue and cloudless. The mountains in the distance are stacked up. Away from the hustle and bustle of city life. "

Figure 7: **Qualitative results of Video-GPT on text-to video-generation.**

`clean_input` and `noised_input` layers Xiao et al. (2024) to transform VAE-encoded image latents into tokens, as well as a `noised_output` layer to convert output tokens back into image latents, with these components having significantly fewer parameters than the main transformer. During the pretraining phase, we employ 320 H20 GPUs and implement a progressive training strategy as shown in Tab. 1. More pretraining and fine-tuning settings can be found in Sup. B.

## 4.2 PRETRAINED MODEL EVALUATION

We evaluate pretrained Video-GPT performance on Physics-IQ Benchmark Motamed et al. (2025) and Kinetics-600 Carreira et al. (2018), which are used to evaluate the ability of models in predicting deterministic and uncertain videos respectively. We set the historical conditioned classifier-free guidance scale $c$=3. Unless specified all subsequent experiments keeps the same setting.

**Physics-IQ Benchmark.** Physics-IQ Benchmark presents a short video clip (3 seconds) depicting real-world physical motion and asks the model to predict future frames (5 seconds). Since macroscopic physical laws typically exhibit deterministic behavior, we used the Physics-IQ Benchmark to assess the model's ability to predict deterministic future events. As shown in Tab. 2, Video-GPT has a superior performance in modeling the physical world (more than **5 points** higher than the 2nd). In addition to Video-GPT, the results of LVM Bai et al. (2023), Open-Sora-Plan v1.3 Lin et al. (2024), and Seine Chen et al. (2023b) are also obtained by our own testing. This result shows that the model based on next clip diffusion pretraining paradigm can better learn world knowledge from video data.

**Kinetics-600.** Kinetics-600 contains 0.3M human motion videos, and human motion is often known for being unpredictable. Therefore, we measure the distance Unterthiner et al. (2018) between model-generated videos and videos from Kinetics-600 to evaluate the model's capability to predict highly uncertain future events, requiring the model to predict 13 future frames based on the 3 given frames. As shown in Tab. 3, Video-GPT achieves the best FVD among all models with a vanilla transformer architecture, rather than the more popular U-Net Ronneberger et al. (2015) or DiT Peebles & Xie (2022), which fully demonstrates the effectiveness of our pretraining.

**Visualization.** As shown in Fig. 5, Video-GPT accurately predicts the water filling and machine lifting physics, while other methods Bai et al. (2023); Chen et al. (2023b) show obvious errors.

## 4.3 ABLATION STUDY

We ablate the properties of Video-GPT pretraining and inference in the Physics-IQ Benchmark. Unless specified, we perform ablation experiments based on the Video-GPT Stage 3 in Tab. 1.

Table 6: **Class to video quantitative comparison on the UCF-101.**

| Model | Resolution | Architecture | VAE | FVD↓ |
|---|---|---|---|---|
| CogVideo Hong et al. (2022) | $160 \times 160$ | Dual-channel Transformer | 2D | 626 |
| Latte Ma et al. (2024) | $256 \times 256$ | DiT | 2D | 478 |
| TATS Ge et al. (2022) | $128 \times 128$ | Time-Sensitive Transformer | - | 332 |
| OmniTokenizer Wang et al. (2024b) | $256 \times 256$ | Vanilla Transformer | 3D | 191 |
| VideoFusion Luo et al. (2023) | $128 \times 128$ | U-Net | - | 173 |
| MAGVITv2-AR Yu et al. (2024) | - | Vanilla Transformer | 3D | 109 |
| ACDIT Hu et al. (2024) | - | DiT | 2D | 90 |
| Make-A-Video Singer et al. (2022) | $256 \times 256$ | U-Net | 2D | 81 |
| MAGVITv2 Yu et al. (2024) | - | Vanilla Transformer | 3D | 58 |
| LARP Wang et al. (2024a) | $128 \times 128$ | Vanilla Transformer | 3D | 57 |
| FAR Gu et al. (2025) | $128 \times 128$ | DiT | 2D | 57 |
| Video-GPT (from scratch) | $240 \times 320$ | Vanilla Transformer | 2D | 489 |
| **Video-GPT** (finetune) | $\mathbf{240 \times 320}$ | **Vanilla Transformer** | **2D** | **53** |

Table 7: Video classification linear probe on UCF-101.

| Method | Top-1 |
|---|---|
| MemDPC Han et al. (2020) | 54.1 |
| VideoMAEv2 Wang et al. (2023a) | 56.4 |
| **Video-GPT** | **58.9** |

Table 8: Video retrieval zero-shot on the MSR-VTT.

| Method | R@1 |
|---|---|
| VideoCLIP Xu et al. (2021) | 10.4 |
| SupportSet Patrick et al. (2020) | 12.7 |
| FiT Bain et al. (2021) | 18.8 |
| **Video-GPT** | **22.8** |

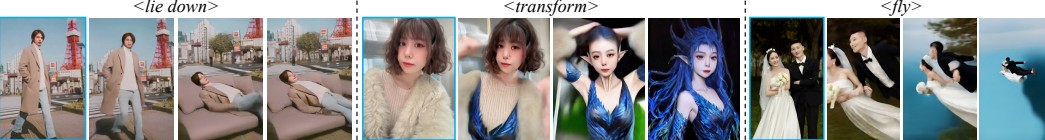

Figure 8: **Qualitative results of Video-GPT on *lie down*, *transform* and *fly* image animation.** The frame in the blue box is the prefix frame we are given as history condition.

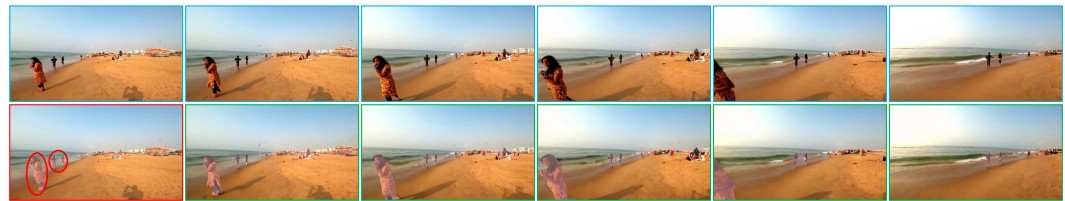

Figure 9: **Qualitative results of Video-GPT on video object segmentation.** The frames in blue box are condition frames. The frame in red box is the 1-st frame of the condition with object mask, and we circle the segmented object in red. Frames in green box are the segmentation result.

**Number of Frames in Video Clip for Inference.** As shown in Tab. 4, within a certain range, as the number of frames $N_k$ processed in parallel for inference within each video clip increases, the quality of the videos generated by the model improves significantly. This thoroughly validates the superiority of our proposed next clip generation paradigm for video generation tasks.

**History Conditioned Classifier-Free Guidance Scale for Inference.** We conduct experiments based on $N_k$=16. As shown in Tab. 4, the Stage 4 model shows the strongest performance when we adjust the scale of history conditioned classifier-free guidance Ho (2022) $c$=3.0.

**Number of Frames for PreTraining.** As shown in Tab. 5, the results show that as the number of pretraining frames increases, the videos generated by Video-GPT are more and more in line with physical laws. This reveals that as the time window of model pretraining increases, the model's modeling of world knowledge is getting better and better.

**Add Noise to Clean Clip for Pretraining.** During inference, there is a deviation between **DNS** and **CL**. To fill this deviation, we add slight noise to clean frames during training as

$$\Phi(k, i) = (\beta + \gamma_{k,i})\Phi(k, i) + (1 - \beta - \gamma_{k,i})\epsilon_{k,i}, \tag{6}$$

where $\beta$ and $\gamma_{k,i} \sim$ Uniform[0,1] refer to the basic retention degree and random retention degree of the clean frame respectively and $\epsilon_{k,i}$ is sampled by $\epsilon_{k,i} \sim \mathcal{N}(\mathbf{0}, \mathbf{I})$. We set $\beta$=0.9. As shown in Tab. 5, the result shows that adding noise to the clean clip during training improves model performance.

**Dataset Scale for Pretraining.** We conduct ablation study on the impact of dataset scale based on OpenVid-1M and Panda-70M. As shown in Tab. 5, the results demonstrate that after expanding the scale of videos for pretraining, Video-GPT's modeling capability for the world significantly improved. This further indicates that Video-GPT has substantial room for improvement, as its pretraining can utilize almost all video data on the internet without annotations, similar to GPT.

## 4.4 DOWNSTREAM TASK EVALUATION

We fine-tuned Video-GPT on downstream tasks, it demonstrates that our model acquired excellent prior knowledge and modal representations through large-scale self-supervised pretraining with videos. Unless specified, we perform experiments based on the Stage 3 pretraining model in Tab. 1.

**Class-to-Video Generation.** As shown in Tab. 6, Video-GPT achieves the state-of-the-art performance on UCF-101 at high resolution. without the better 3D VAE. In addition, we test the Video-GPT trained from scratch on UCF-101, and the results show that our pretraining plays a very important role in the downstream task of class-to-video. Qualitative results are presented in Fig. 6.

**Text-to-Video Generation.** We employ the same text encoder as SDv2 Rombach et al. (2021). We filter OpenVid-1M based on motion and aesthetic scores Lin et al. (2024) and obtain 0.3M pairs for training, which is a relatively small dataset. As shown in Fig. 7, it demonstrates that our Video-GPT can achieve effective cross-modal generation with only a limited amount of text-video pairs.

**Image Animation.** In Fig. 8, we fine-tune Video-GPT 2K steps on three training sets with no more than 100 videos each, it shows amazing generalization ability on test cases outside the training set.

**Video Classification.** As shown in Tab. 7, Video-GPT linear probe on UCF-101 surpasses Video-MAEv2 Wang et al. (2023a), a pretrained model commonly used in the field of video understanding.

**Video Retrieval.** In Tab. 8, Video-GPT achieves impressive zero-shot performance on the MSR-VTT Xu et al. (2016) and surpasses VideoClip Xu et al. (2021) which was trained from scratch.

**Video Object Segmentation.** We finetune Video-GPT 5K steps on GetIn-1M 1K subset. In Fig. 9, it alse present good generalization performance when transferred to low level understanding task.

## 5 CONCLUSION

We introduce Video-GPT, a concise yet powerful large video foundation model that unites autoregressive modeling with diffusion through next clip diffusion. Treating each video clip as a word token allows the model to inherit the self-supervised property of GPT while retaining the sharp synthesis quality of diffusion. Pretraining on 70M unlabeled videos yields state-of-the-art accuracy on deterministic Physics-IQ Benchmark and uncertain Kinetics-600 forecasting, and fine-tuning transfers seamlessly to six challenging and diverse downstream tasks including generation and understanding. Future work will explore multi-modal pretraining, reinforcement-driven world interaction.

ACKNOWLEDGMENTS

This work was supported by Guangdong Science and Technology Program (Grant No. 2024TQ08X365).

ETHICS STATEMENT

Our Video-GPT is a concise yet powerful large video foundation model that unites autore- gressive modeling with diffusion through next clip diffusion. To ensure ethical compliance, our training data is carefully selected from public sources and take deliberate measures to minimize potential biases, fully aligning with universal ethical guidelines. We explicitly emphasize that this framework is not intended for misuse in achieving harmful purposes; downstream users are encouraged to adhere to ethical principles when applying the technology. Additionally, all authors declare no conflicts of interest related to this work.

REPEATABILITY STATEMENT

To ensure full reproducibility, we will publicly release all code and data necessary to replicate our experiments. Comprehensive implementation details, including model architecture, hyperparameters, and training methodology, are provided in this paper and its appendix. We are committed to open-sourcing all essential resources to ensure that our findings can be fully verified and built upon by the research community.

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

---

**Algorithm 1** Video-GPT Pre-training

---

1: **Input:**
2: $V$: The set of all training videos.
3: $M_\theta$: Video-GPT model with parameters $\theta$.
4: $\text{VAE}_E$: The pre-trained VAE Encoder.
5: $T$: Total number of training steps.

6: **Main Training Loop:**
7: **for** $step = 1 \to T$ **do**
8:     *// 1. Input Sequence Construction (Sec. 3.1)*
9:     $v \leftarrow \text{sample\_video}(V)$                              ▷ Sample a video from the training set
10:     $\text{all\_frames} \leftarrow \text{sample\_n\_frames}(v)$              ▷ Sample $n$ frames from the video
11:     $\text{clips} \leftarrow \text{partition\_into\_k\_clips}(\text{all\_frames})$         ▷ $K$ is randomly sampled

12:     $S_{\text{input}} \leftarrow [\,]$                             ▷ Initialize full input sequence for the model
13:     $S_{\text{target}} \leftarrow [\,]$                          ▷ Initialize ground truth clean latents for loss

14:     **for** $k = 1 \to \text{length}(\text{clips})$ **do**
15:         $C_k \leftarrow \text{clips}[k]$
16:         $\Phi_k \leftarrow \text{VAE}_E(C_k)$                    ▷ Encode clean clip to latent space $\Phi_k$

        *// (Optional) Add slight noise for training robustness (Sec. 4.3, Eq. 6)*
17:         $\Phi_{k,\text{noisy}} \leftarrow \text{add\_slight\_noise}(\Phi_k)$

        *// Create a noisy version via forward diffusion (flow matching)*
18:         $\alpha_k \leftarrow \text{sample\_from\_uniform}(0, 1)$        ▷ Sample a single noise level for the clip
19:         $\varepsilon_k \sim \mathcal{N}(0, I)$ with shape of $\Phi_k$        ▷ Sample from a standard Gaussian
20:         $\Psi_k \leftarrow \alpha_k \Phi_k + (1 - \alpha_k)\varepsilon_k$             ▷ Noisy latent features

        *// Format clips into token sequences*
21:         $\text{NS}_k \leftarrow \text{format\_noisy\_sequence}(\Psi_k, \alpha_k)$     ▷ e.g., $[<\text{diff}>, \alpha_k, \Psi_{k,1}, \dots]$
22:         $\text{CL}_k \leftarrow \text{format\_clean\_sequence}(\Phi_{k,\text{noisy}})$    ▷ e.g., $[<\text{img}>, \Phi_{k,1}, <\backslash\text{img}>, \dots]$

23:         Append $\text{NS}_k$ and $\text{CL}_k$ to $S_{\text{input}}$
24:         Append $\Phi_k$ to $S_{\text{target}}$            ▷ Target for loss is the original clean latent
25:     **end for**

26:     *// 2. Pre-training with Next Clip Diffusion (Sec. 3.2)*
        *// Hierarchical attention mask is applied internally by $M_\theta$*
27:     $\text{predicted\_}\Phi \leftarrow M_\theta(S_{\text{input}})$                ▷ Model forward pass

28:     *// 3. Loss Calculation and Optimization*
        *// L2 loss is calculated only on outputs corresponding to noisy inputs*
29:     $\mathcal{L} \leftarrow \text{l2\_loss}(\text{predicted\_}\Phi, S_{\text{target}})$
30:     $\mathcal{L}.\text{backward}()$
31:     $\text{optimizer.step}()$
32:     $\text{optimizer.zero\_grad}()$
33: **end for**

---

# A   PSEUDOCODE OF VIDEO-GPT

## A.1   TRAINING PSEUDOCODE

As shown in Algorithm 1.

---

**Algorithm 2** Video-GPT Autoregressive Inference

---

1: **Input:**
2: $M_\theta$: Pre-trained Video-GPT model.
3: $\text{VAE}_E, \text{VAE}_D$: VAE Encoder and Decoder.
4: $C_1$: Initial conditioning video clip (the first clip).
5: $N$: Total number of clips to generate.
6: $c$: Classifier-Free Guidance scale (Sec. 4.3).
7: $T$: Total number of denoising steps.

8: *// 1. Initialization* GenerateVideo($C_1, N, c, T$)
9: $DNS_1 \leftarrow \text{VAE}_E(C_1)$         $\triangleright$ Encode the initial clip to a Denoised Sequence.
10: $Generated\_Latents \leftarrow [DNS_1]$       $\triangleright$ Initialize list of generated latent clips.

11: *// 2. Autoregressive Generation Loop (Sec. 3.3)*
12: **for** $k = 1$ **to** $N - 1$ **do**
13:     $NS_{\text{next}} \leftarrow \text{sample\_from\_gaussian}(\text{shape}(DNS_1))$    $\triangleright$ Prepare noisy input for clip $k + 1$.

14:     *// 3. T steps denoising*
15:     **for** $t = 1$ **to** $T$ **do**
16:        $Context \leftarrow Generated\_Latents$        $\triangleright$ Clean latents as context.

17:        *// 4. Denoise using History Conditioned Classifier-Free Guidance*
18:        $input_{\text{cond}} \leftarrow \text{create\_input\_sequence}(Context, NS_{\text{next}})$
19:        $pred_{\text{cond}} \leftarrow M_\theta(input_{\text{cond}})$      $\triangleright$ Conditional prediction (with history).
20:        $input_{\text{uncond}} \leftarrow \text{create\_input\_sequence}(\emptyset, NS_{\text{next}})$
21:        $pred_{\text{uncond}} \leftarrow M_\theta(input_{\text{uncond}})$     $\triangleright$ Unconditional prediction (null context).
22:        $cfg_{\text{cond}} \leftarrow pred_{\text{uncond}} + c \cdot (pred_{\text{cond}} - pred_{\text{uncond}})$     $\triangleright$ Predictions with CFG.
23:        $NS_{\text{next}} \leftarrow \text{Flow\_Matching}(cfg_{\text{cond}}, t)$    $\triangleright$ Update noisy latent via flow matching.
24:     **end for**
25:     APPEND($Generated\_Latents, NS_{\text{next}}$)    $\triangleright$ Append the newly generated clean latent clip.
26: **end for**

27: *// 5. Decode the final video*
28: $Final\_Video\_Clips \leftarrow []$
29: **for each** $latent\_clip$ **in** $Generated\_Latents$ **do**
30:    $video\_clip \leftarrow \text{VAE}_D(latent\_clip)$
31:    APPEND($Final\_Video\_Clips, video\_clip$)
32: **end for**
33: $Final\_Video \leftarrow \text{concatenate}(Final\_Video\_Clips)$
34: **return** $Final\_Video$

---

## A.2 INFERENCE PSEUDOCODE

As shown in Algorithm 2.

## A.3 UNIFORM SAMPLING PSEUDOCODE

As shown in Algorithm 3.

## B MORE IMPLEMENTATION DETAILS

As shown in Tab. 9, 10, 11, 12, 13, 14, 15, 16, 17 and 18.

## C MORE VISUALIZATION RESULTS

The full mask of the input in Fig. 2 (a) is shown in Fig. 10.

Table 9: **Pretraining Stage 1 setting.**

| config | Panda-70M |
|---|---|
| optimizer | AdamW Loshchilov & Hutter (2017) |
| optimizer momentum | $\beta_1, \beta_2$=0.9, 0.95 |
| weight decay | 0.1 |
| learning rate schedule | consistent |
| learning rate | 1e-4 |
| warmup steps Goyal et al. (2017) | 1000 |
| total steps | 300000 |
| input frame | 16 |
| resulution | flexible |
| longest side | 320 |
| frame interval | 4 |
| video clip | 16 |
| computing unit | 320 |

Table 10: **Pretraining Stage 2 setting.**

| config | Panda-70M |
|---|---|
| optimizer | AdamW Loshchilov & Hutter (2017) |
| optimizer momentum | $\beta_1, \beta_2$=0.9, 0.95 |
| weight decay | 0.1 |
| learning rate schedule | consistent |
| learning rate | 1e-4 |
| warmup steps Goyal et al. (2017) | 1000 |
| total steps | 25000 |
| input frame | 48 |
| resulution | flexible |
| longest side | 320 |
| frame interval | 4 |
| video clip | $\sim \text{Uniform}\{2, 3, \cdots, 48\}$ |
| computing unit | 320 |

Table 11: **Pretraining Stage 3 setting.**

| config | Panda-70M |
|---|---|
| optimizer | AdamW Loshchilov & Hutter (2017) |
| optimizer momentum | $\beta_1, \beta_2$=0.9, 0.95 |
| weight decay | 0.1 |
| learning rate schedule | consistent |
| learning rate | 1e-4 |
| warmup steps Goyal et al. (2017) | 1000 |
| total steps | 40000 |
| input frame | 48 |
| resulution | flexible |
| longest side | 320 |
| frame interval | $\sim \text{Uniform}\{4, 5, \cdots, 12\}$ |
| video clip | $\sim \text{Uniform}\{2, 3, \cdots, 48\}$ |
| computing unit | 320 |

Table 12: **Pretraining Stage 4 setting.**

| config | Panda-70M |
|---|---|
| optimizer | AdamW Loshchilov & Hutter (2017) |
| optimizer momentum | $\beta_1, \beta_2$=0.9, 0.95 |
| weight decay | 0.1 |
| learning rate schedule | consistent |
| learning rate | 2e-5 |
| warmup steps Goyal et al. (2017) | 1000 |
| total steps | 20000 |
| input frame | 80 |
| resulution | flexible |
| longest side | 320 |
| frame interval | $\sim \text{Uniform}\{4, 5, \cdots, 12\}$ |
| video clip | $\sim \text{Uniform}\{2, 3, \cdots, 80\}$ |
| computing unit | 320 |

Table 13: **Class to video fine-tuning on UCF-101 setting.**

| config | UCF-101 |
|---|---|
| optimizer | AdamW Loshchilov & Hutter (2017) |
| optimizer momentum | $\beta_1, \beta_2$=0.9, 0.95 |
| weight decay | 0.1 |
| learning rate schedule | consistent |
| learning rate | 1e-4 |
| warmup steps Goyal et al. (2017) | 1000 |
| total steps | 165000 |
| input frame | 16 |
| resulution | 240×320 |
| longest side | 320 |
| frame interval | 1 |
| video clip | 1 |
| computing unit | 64 |

Table 14: **Text to video fine-tuning on OpenVid-0.3M setting.**

| config | OpenVid-0.3M |
|---|---|
| optimizer | AdamW Loshchilov & Hutter (2017) |
| optimizer momentum | $\beta_1, \beta_2$=0.9, 0.95 |
| weight decay | 0.1 |
| learning rate schedule | consistent |
| learning rate | 1e-4 |
| warmup steps Goyal et al. (2017) | 1000 |
| total steps | 70000 |
| input frame | 24 |
| resulution | flexible |
| longest side | 320 |
| frame interval | 4 |
| video clip | 1 |
| computing unit | 128 |

Table 15: **Image animation fine-tuning on animation dataset setting.**

| config | Anim. Dataset |
|---|---|
| optimizer | AdamW Loshchilov & Hutter (2017) |
| optimizer momentum | $\beta_1, \beta_2$=0.9, 0.95 |
| weight decay | 0.1 |
| learning rate schedule | consistent |
| learning rate | 1e-4 |
| warmup steps Goyal et al. (2017) | 1000 |
| total steps | 2000 |
| input frame | 48 |
| resulution | flexible |
| longest side | 320 |
| frame interval | flexible |
| video clip | 2 |
| computing unit | 8 |

Table 16: **Video classification fine-tuning on UCF-101 setting.**

| config | UCF-101 |
|---|---|
| optimizer | AdamW Loshchilov & Hutter (2017) |
| optimizer momentum | $\beta_1, \beta_2$=0.9, 0.95 |
| weight decay | 0.1 |
| learning rate schedule | consistent |
| learning rate | 1e-4 |
| warmup steps Goyal et al. (2017) | 1000 |
| total steps | 10000 |
| input frame | 16 |
| resulution | 224×224 |
| longest side | 224 |
| frame interval | 1 |
| video clip | 1 |
| computing unit | 8 |

Table 17: **Video retrieval fine-tuning on Panda-70M.**

| config | Panda-70M |
|---|---|
| optimizer | AdamW Loshchilov & Hutter (2017) |
| optimizer momentum | $\beta_1, \beta_2$=0.9, 0.95 |
| weight decay | 0.1 |
| learning rate schedule | consistent |
| learning rate | 1e-4 |
| warmup steps Goyal et al. (2017) | 1000 |
| total steps | 200000 |
| input frame | 8 |
| resulution | 224×224 |
| longest side | 224 |
| frame interval | flexible |
| video clip | 1 |
| computing unit | 64 |

Table 18: **Video object segmenration fine-tuning on GetIn-1K setting.**

| config | GetIn-1K |
|---|---|
| optimizer | AdamW Loshchilov & Hutter (2017) |
| optimizer momentum | $\beta_1, \beta_2$=0.9, 0.95 |
| weight decay | 0.1 |
| learning rate schedule | consistent |
| learning rate | 1e-4 |
| warmup steps Goyal et al. (2017) | 1000 |
| total steps | 5000 |
| input frame | 48 |
| resulution | flexible |
| longest side | 224 |
| frame interval | flexible |
| video clip | 2 |
| computing unit | 8 |

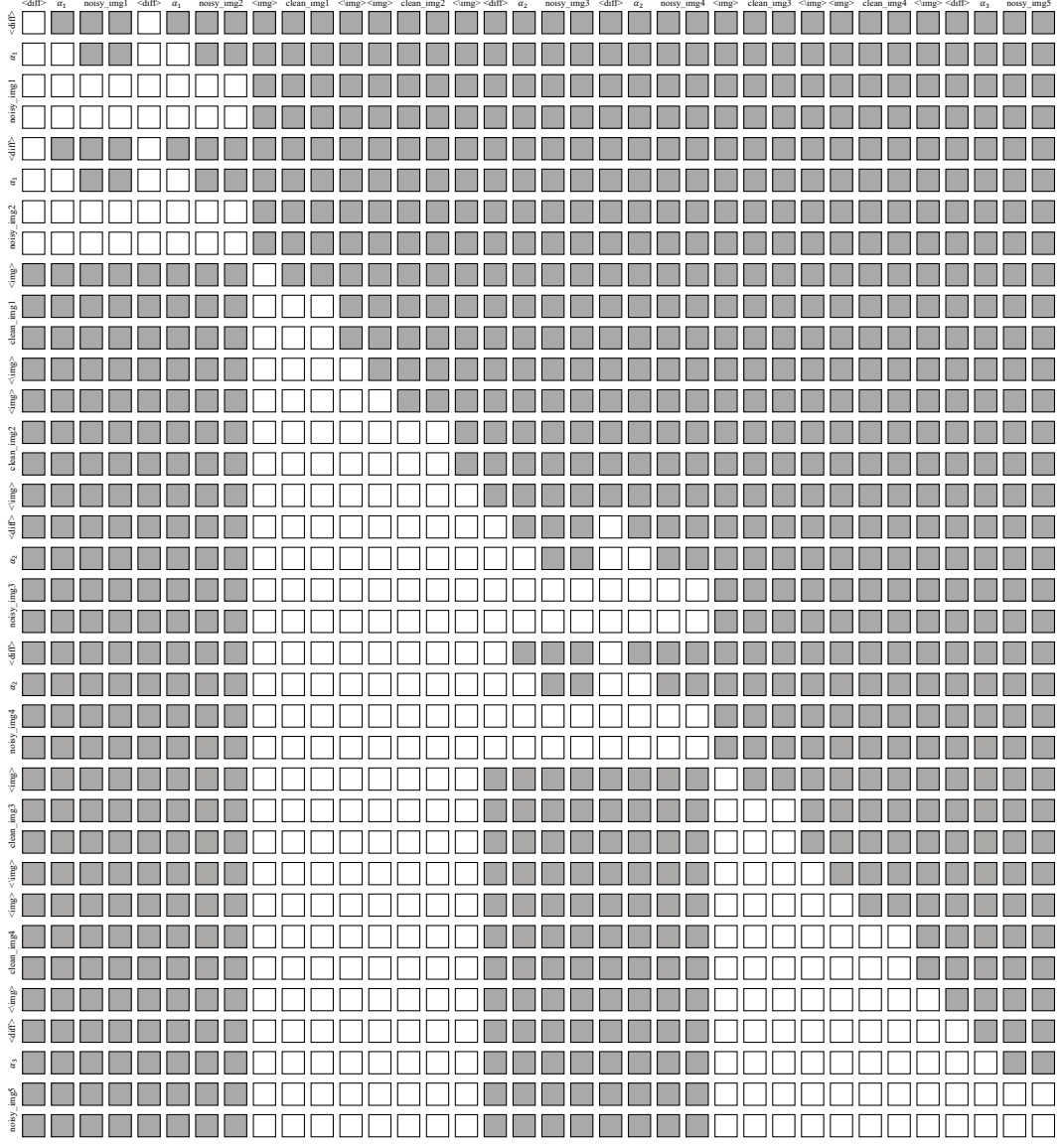

Figure 10: **The full attention mask of Fig. 2 (a) training input.**

---

**Algorithm 3** GenerateRandomList

---

1: **Input:**
2: $num\_frames$: Total number of frames (integer, $\geq 1$).
3: **Output:**
4: $parts$: A list of positive integers that sum to $num\_frames$.

5: *// 1. Initialization*
6: $parts \leftarrow []$      ▷ Initialize the output list.
7: $previous \leftarrow 0$      ▷ Track the last cut position (start at 0).

8: *// 2. Decide the number of segments internally (no extra input)*
9: $k \leftarrow$ random_integer$(1, num\_frames)$      ▷ Number of parts; chosen uniformly to mimic sampling logic.
10: $m \leftarrow k - 1$      ▷ Number of cut points to sample; $m = 0$ yields no cuts.

11: *// 3. Sample and sort cut positions from the valid range*
12: $cuts \leftarrow$ sorted(random_sample(range$(1, num\_frames), m$))      ▷ Distinct cuts in $\{1, \ldots, num\_frames - 1\}$; sorted ascending.

13: *// 4. Build parts by consecutive differences*
14: **for each** $cut$ **in** $cuts$ **do**
15:      APPEND$(parts, cut - previous)$      ▷ Length of the segment between $previous$ and $cut$.
16:      $previous \leftarrow cut$      ▷ Advance the cursor to the current cut.
17: **end for**
18: APPEND$(parts, num\_frames - previous)$      ▷ Append the final segment from the last cut to $num\_frames$.

19: **return** $parts$

---

We show the results of Video-GPT for zero-shot long video prediction in Fig. 11 and 12.

In order to demonstrate the capabilities of Video-GPT more comprehensively, we test Video-GPT's prediction ability for videos with partially static clips (minimal motion) and partially with abrupt motion. As can be seen from Fig. 13, Video-GPT is able to handle such extreme video cases. Although there is still much room for improvement, we believe that Video-GPT has indeed learned highly generalizable world knowledge from massive amounts of video data.

As can be seen from Fig. 14, we employ Video-GPT prediction to generate a 1-minute long video. Experiments have shown that Video-GPT can generate videos up to 1-minute long while maintaining temporal coherence and content consistency. As shown in Fig. 15, we generate 1-minute videos using Open-Sora-Plan Lin et al. (2024). From the comparison between Fig. 14 and 15, we can find that the final images of the video generated by Open-Sora-Plan has been significantly collapsed, while the image generated by Video-GPT is still relatively clear. This comparison once again verifies the strong temporal robustness of our Video-GPT.

## D   LIMITATIONS

Our model exhibits remarkable performance and generalization across diverse tasks, affirming the strength of the next clip diffusion paradigm. It is important to note, however, that our experiments are conducted with a moderately scaled model due to available computational resources. This choice, while pragmatic, has not detracted from the quality of our good results. Looking forward, further scaling may reveal additional performance gains. Nonetheless, the current configuration already sets a high benchmark, underscoring both the efficacy and robustness of our approach in video generative modeling.

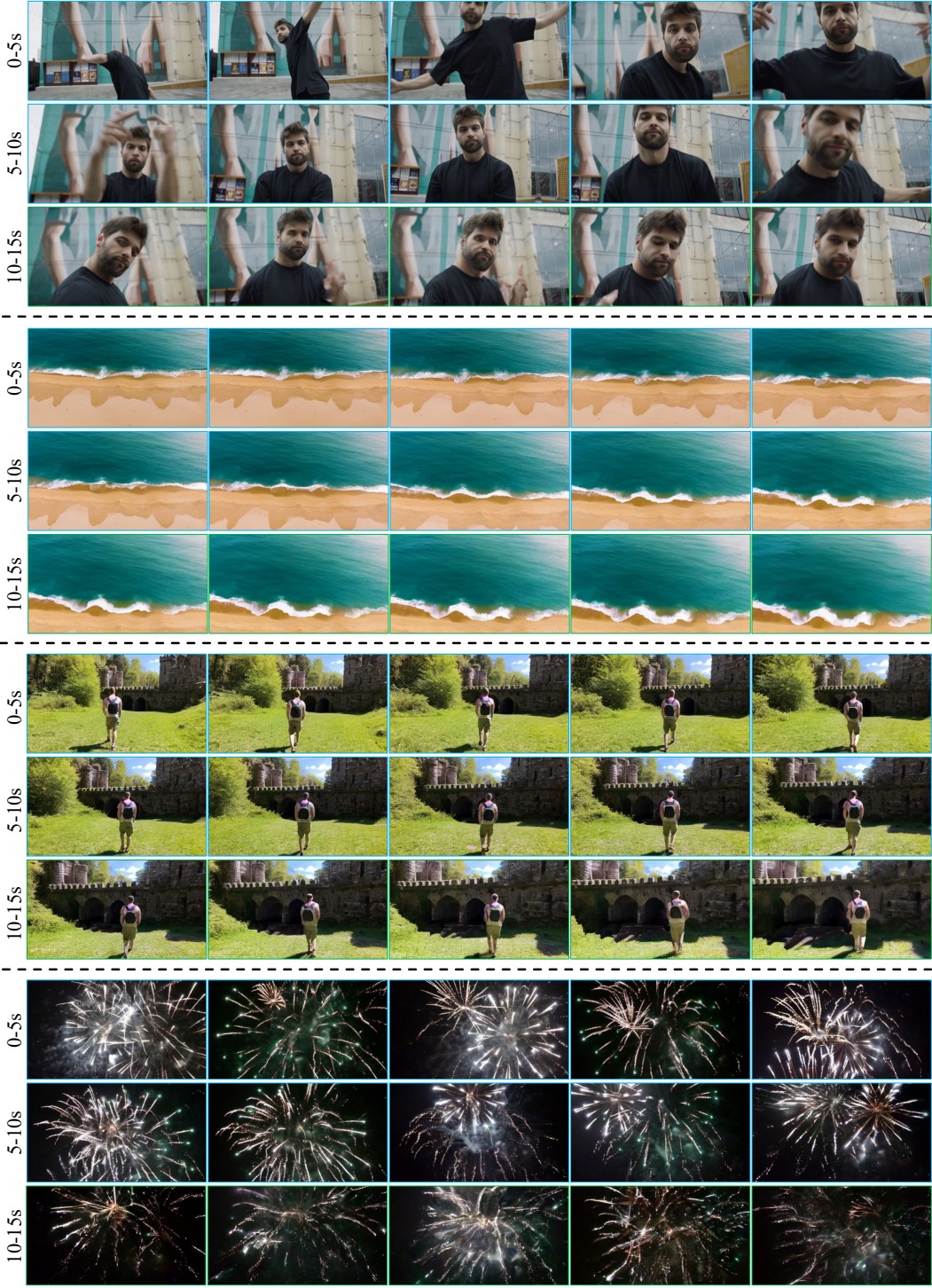

Figure 11: **The long video zero-shot prediction result generated by Video-GPT.** The frames in blue box are condition frames. Frames in green box are the generated prediction result.

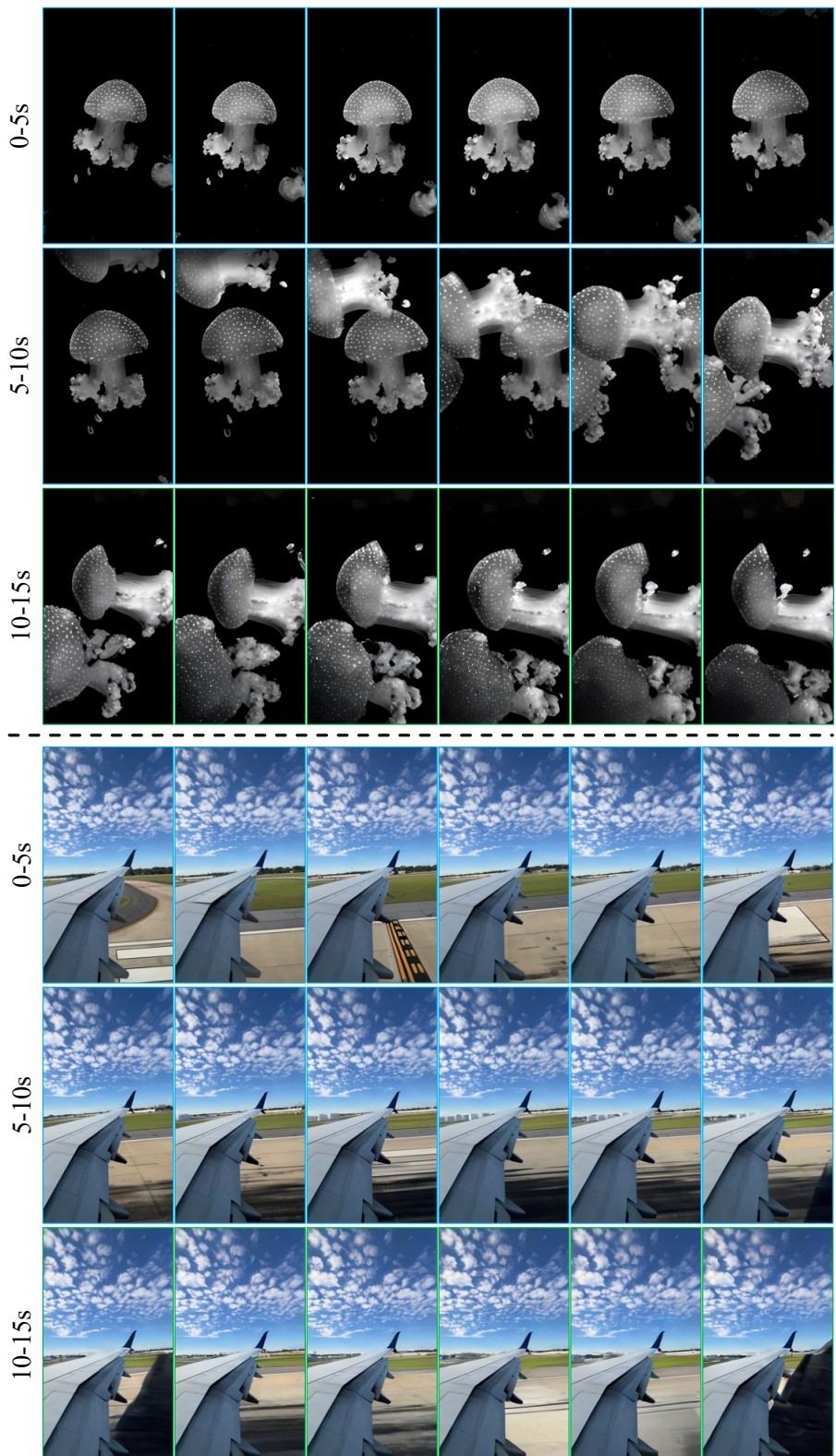

Figure 12: **The long video zero-shot prediction result generated by Video-GPT.** The frames in blue box are condition frames. Frames in green box are the generated prediction result.

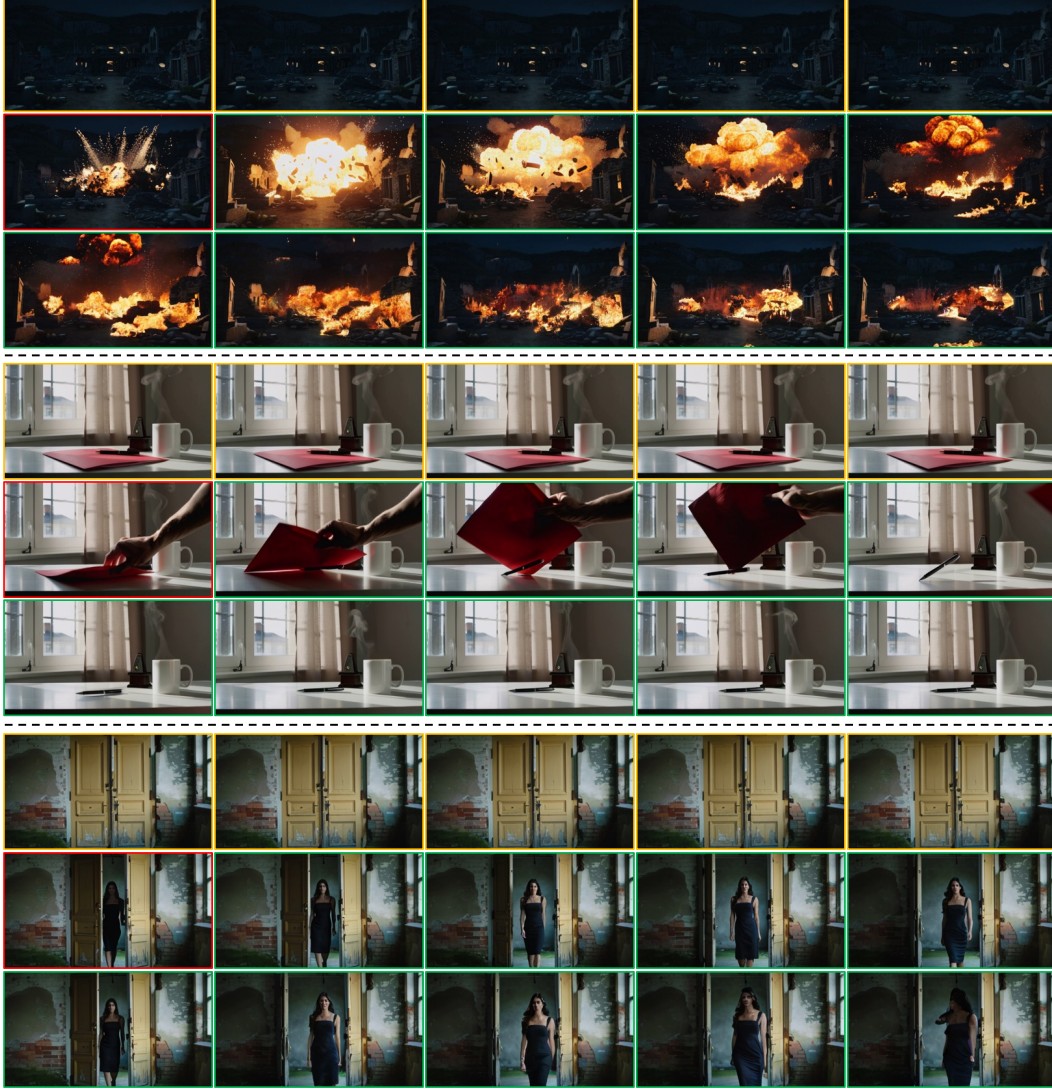

Figure 13: **The partly stationary clips (minimal motion) and partly abrupt motion video zero-shot prediction result generated by Video-GPT.**
To test the model's ability to handle extreme cases with partly stationary clips (minimal motion) and partly abrupt motion, we copied an image 11 times to simulate a static video, and then edited the image using FLUX-Kontext Labs et al. (2025). The edited image was then used as the 12th frame of the video to simulate a abrupt, large motion, and Video-GPT (stage4) predicted the next 48 frames. The frames in the orange boxes are given stationary video frames, the red box is abrupt motion frame, and the green boxes are video frames generated by Video-GPT prediction.

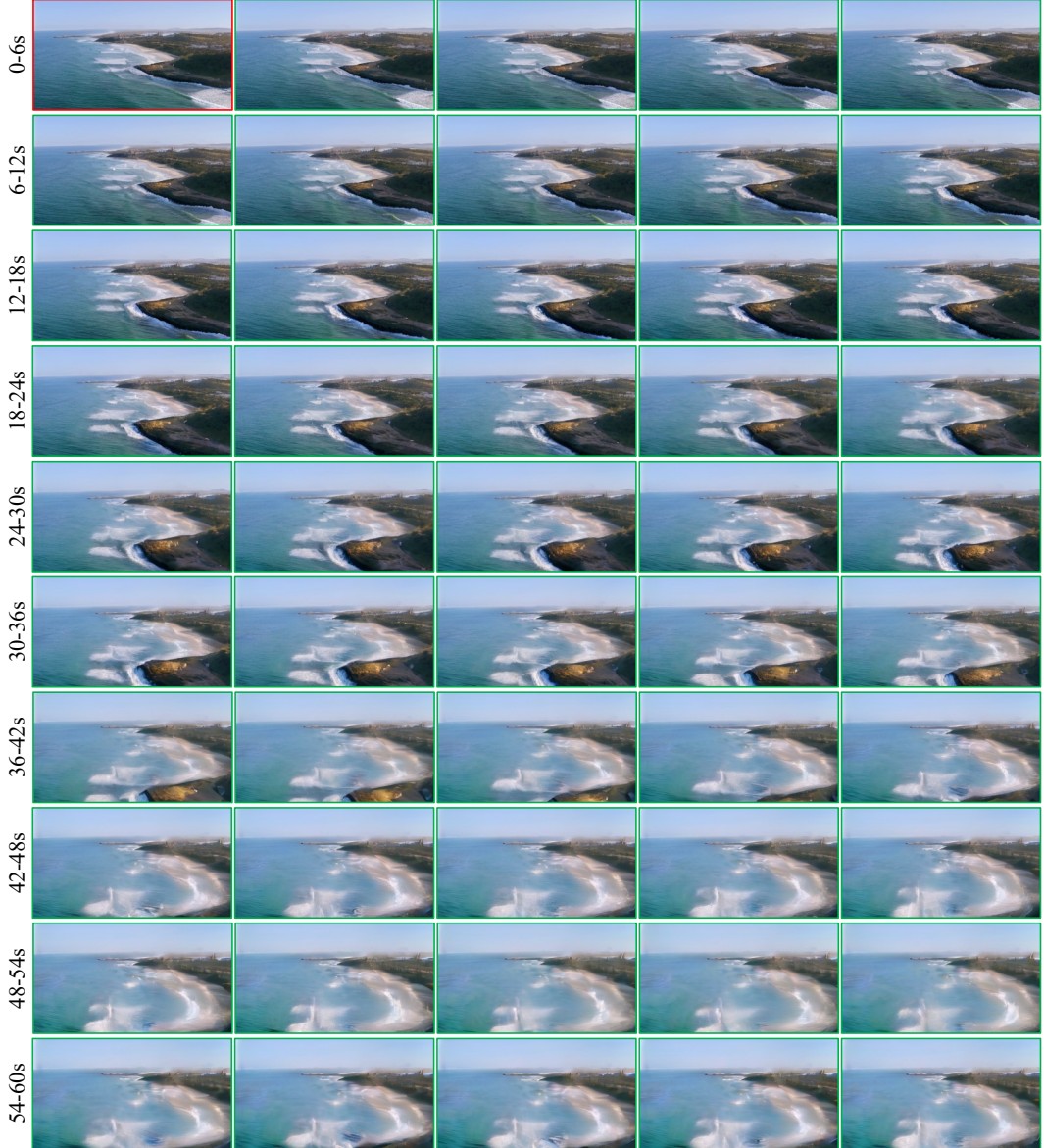

Figure 14: **1-minute video generated by Video-GPT.**
The frame in the red boxes is conditional video frame, the green boxes are video frames generated by Video-GPT prediction.

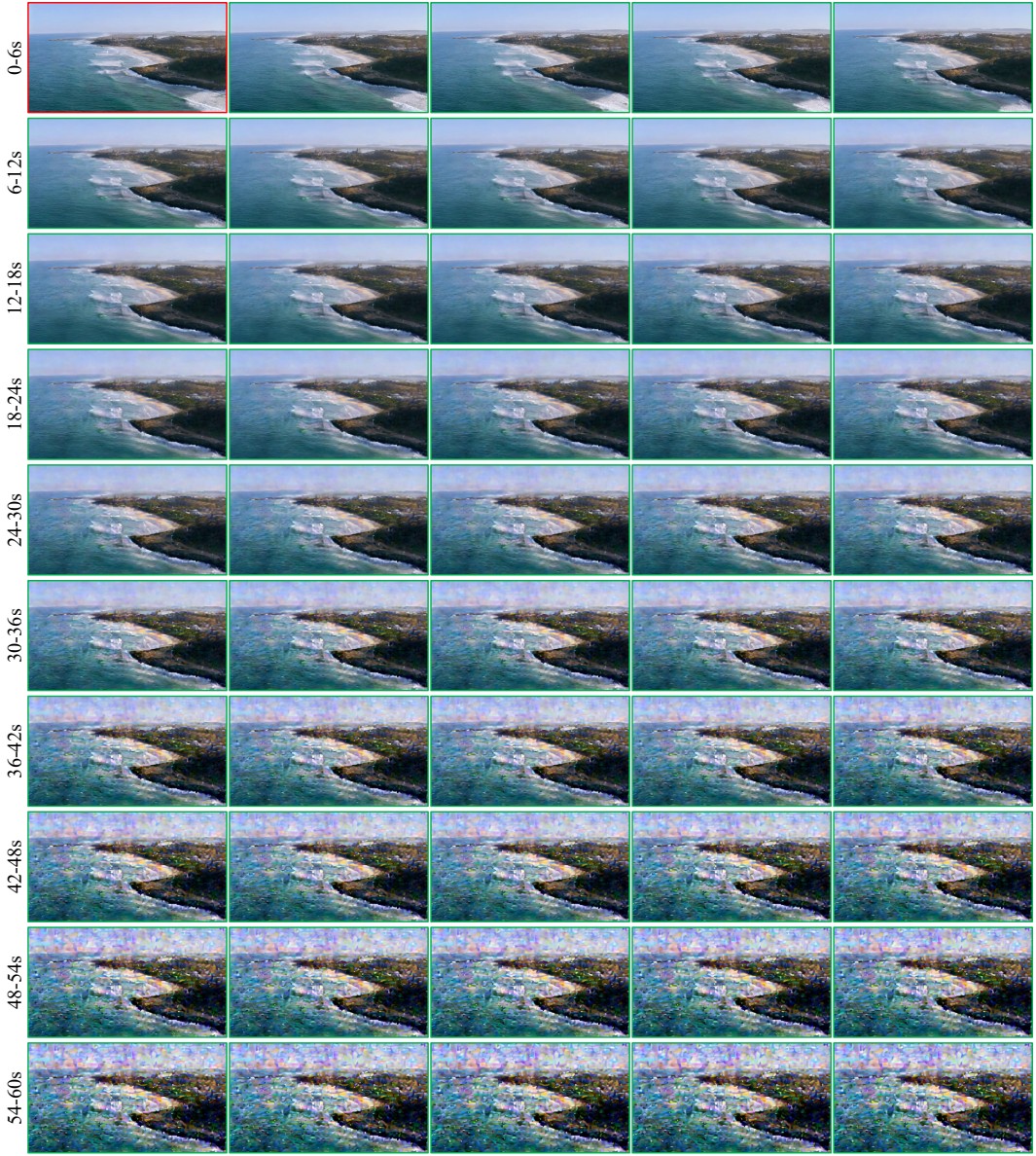

Figure 15: **1-minute video generated by Open-Sora-Plan Lin et al. (2024).**
The frame in the red box is conditional video frame, the green boxes are video frames predicted by Open-Sora-Plan.

# E   DECLARATION OF USE OF LARGE LANGUAGE MODELS (LLM)

We affirm that this paper was primarily written by the authors. Large Language Models (LLMs) were utilized solely as general-purpose assistive tools for language refinement, grammar correction, and stylistic improvements during the writing process. Specifically, Gemini 2.5 Flash (DeepMind, 2025) was employed for minor text polishing and rephrasing to enhance clarity and readability. No LLM was used for conceptual ideation, experimental design, data analysis, or generating any substantive content of the research.

