# OpenReview forum: "Video-GPT via Next Clip Diffusion"
_ICLR.cc/2026/Conference — ICLR 2026 Poster_

### Official Review · Reviewer_wpuq · 2025-10-25

**Soundness:** 3
**Presentation:** 2
**Contribution:** 3
**Rating:** 6
**Confidence:** 3

**Summary:**

This paper introduces Video-GPT, a novel foundation model for video generation and understanding, built upon an elegant analogy of treating video as a new language. The core contribution is a "next clip diffusion" pretraining paradigm, which ingeniously combines autoregressive modeling and diffusion. By treating video clips as "visual words," the model autoregressively predicts the next clip by denoising a noisy version, conditioned on the history of previously generated clean clips. This self-supervised approach allows for effective pretraining on large-scale unlabeled video data. The pretrained Video-GPT achieves state-of-the-art performance on the Physics-IQ benchmark, demonstrating a strong capacity for world modeling, and shows excellent generalization across six diverse downstream video generation and understanding tasks.

**Strengths:**

1. The proposed "next clip diffusion" paradigm is a novel and insightful method for unifying autoregressive and diffusion models for video. The concept of conditioning the denoising of a future clip on the clean history of past clips is a clever and distinct approach that effectively leverages the strengths of both modeling families for long-term video prediction.
2. The paper presents exceptionally strong empirical results. Achieving a state-of-the-art score of 34.97 on the Physics-IQ benchmark, significantly outperforming prior work, is a standout achievement that validates the model's ability to learn physical dynamics. Furthermore, the model's strong performance across a wide array of 6 downstream tasks (including generation and understanding) underscores its quality and versatility as a powerful video foundation model. The ablation studies are thorough and convincingly support the main design choices.

**Weaknesses:**

1. The proposed input formulation, an interleaved sequence of noisy and clean clips [NS(1), CL(1), ..., NS(K), CL(K)], effectively doubles the sequence length processed by the transformer compared to methods that only use historical context. Given the quadratic complexity of attention, this could be a significant computational bottleneck, potentially limiting scalability. A brief analysis of the computational trade-offs would be beneficial.
2. The paper states that frames are divided into K clips, K∼Uniform{2,3,...,N}. This process is central to the method, but its details are sparse. It is unclear if clips are contiguous blocks or formed differently. The impact of this random clip partitioning strategy on training stability and performance is not ablated, yet it seems like a critical hyperparameter.
3. While the paper provides a good overview of related work, the discussion on how "next clip diffusion" specifically differs from other recent hybrid autoregressive-diffusion models for video (e.g., VideoPoet, SEINE) could be more detailed. A deeper comparative analysis would help to better contextualize the novelty of the proposed conditioning and generation scheme

**Questions:**

1. The inference process is autoregressive, where the model's own generated (denoised) clips are used as the clean history for subsequent steps. Have you investigated the model's robustness to error accumulation? For instance, does a minor artifact in a generated clip DNS(k)$ degrade the quality of all future clips?
2. You mention that the model is trained to predict the clean clip directly (x-prediction) rather than the noise (ϵ-prediction) or velocity (v-prediction) to keep the training simple. This is a departure from many modern diffusion frameworks. Could you elaborate on this design choice? Did you experiment with other prediction targets, and did x-prediction yield superior performance?
3. The progressive training strategy in Table 1, which starts with short clips (effectively next-frame prediction) and gradually increases clip length, is interesting. Could you provide more intuition on why this curriculum is effective? Does learning fine-grained temporal dynamics first provide a better foundation for the model before it tackles longer-range dependencies?

---

> ### Author Response · Authors · 2025-11-19
> **Reply to Reviewer wpuq (1/3)**
>
> Dear Reviewer wpuq,
>
> Thank you for your insightful review and for finding our method "**novel and insightful**." We appreciate your detailed questions regarding our design choices.
>
> ---
>
> >**Q1:** The proposed input formulation, an interleaved sequence of noisy and clean clips [NS(1), CL(1), ..., NS(K), CL(K)], effectively doubles the sequence length processed by the transformer compared to methods that only use historical context. Given the quadratic complexity of attention, this could be a significant computational bottleneck, potentially limiting scalability. A brief analysis of the computational trade-offs would be beneficial.
>
> **A1:** This is a crucial question about our core design. Below I will analyze in detail the changes that Interleaved Modeling brings to the training efficiency of Video-GPT.
> 1. **w/ Interleaved Modeling:** When we apply interleaved modeling, Video-GPT can perform denoising training on K clips in a batch at the same time, which means **interleaved_batch_size = batch_size × K**. After our **1,000,000** rounds of testing, when the training frame number is 16, the mean value of K is **9**, and when the training frame number is **80**, the mean value of K is 41, which will greatly improve our training efficiency.
> 2.  **w/o Interleaved Modeling:** If we do not apply interleaved modeling, the length of the token sequence will be reduced by half. When the number of tokens is small, the computational overhead of the model is mainly concentrated in the FFN module, which increases linearly with the number of tokens. At this time, **no_interleaved_batch_size=batch_size × 2**. When the number of tokens is large, the computational overhead of the model is mainly concentrated in the attention part, which increases with the square of the number of tokens. At this time, **no_interleaved_batch_size=batch_size × 4**.
>
> However, better than usual, K is usually greater than 4, which means **interleaved_batch_size > no_interleaved_batch_size**. In other words, the application of interleaved modeling increases the batch size of Video-GPT training at the same time, which greatly improves the training efficiency of Video-GPT.
>
> ---
>
> >**Q2:** The paper states that frames are divided into K clips, K∼Uniform{2,3,...,N}. This process is central to the method, but its details are sparse. It is unclear if clips are contiguous blocks or formed differently. The impact of this random clip partitioning strategy on training stability and performance is not ablated, yet it seems like a critical hyperparameter.
>
> **A2:** Thank you for pointing this out. Below we give the real code for uniform sampling:
>
>     def generate_random_list(num_frames):
>
>         cuts = sorted(random.sample(range(1, num_frames), k-1))
>
>         parts = []
>
>         previous = 0
>
>         for cut in cuts:
>
>             parts.append(cut - previous)
>
>             previous = cut
>
>         parts.append(num_frames - previous)
>
>         return parts
>
> In the related work of revised manuscript, we added pseudocode about uniform sampling in the Appendix A.3 section to facilitate readers' understanding.
>
> In response to the effectiveness of this clip sampling, we also conducted additional ablation experiments. We set it to the same as the frame number, at which point Video-GPT degenerates into next-frame prediction:
> | Frame Num. | Clip Num. | Phy. IQ Score |
> | :--- | :--- | :--- |
> | 16 | 16 | 0.00 |
> | 16 | ~Uniform{2,3,···,16} | 22.06 |
>
> This is a very strange result that has troubled us for quite some time. When the model is pre-trained using next-frame prediction, the model cannot perform correct inference, but after we continue training on the next-clip prediction task, the model's ability seems to suddenly "emerge". We attribute this special phenomenon to the accumulation of errors in the model during next-frame prediction, and the next clip prediction can well alleviate this phenomenon.

---

> ### Author Response · Authors · 2025-11-19
> **Reply to Reviewer wpuq (2/3)**
>
> >**Q3:** While the paper provides a good overview of related work, the discussion on how "next clip diffusion" specifically differs from other recent hybrid autoregressive-diffusion models for video (e.g., VideoPoet, SEINE) could be more detailed. A deeper comparative analysis would help to better contextualize the novelty of the proposed conditioning and generation scheme
>
> **A3:** This is a critical point, and we thank you for pushing for more clarity. Our "next clip diffusion" paradigm differs from prior works in two fundamental ways:
>
> 1. **Self-Supervised Pre-Training:** Unlike Self-forcing or APT2, our Video-GPT is pre-trained through self-supervised pre-training similar to LLM, does not require any annotations other than videos in this process. Through pre-training on the open source Panda-70M data set, we have achieved and surpassed the physical world modeling capabilities of the top closed-source models like Gen3 and Sora.
> 2. **Next Clip Diffusion:** Many prior works are autoregressive at the frame level, our method is autoregressive at the clip level. This is more efficient as it allows parallel, bidirectional processing (diffusion) within a multi-frame clip, but maintains temporal order between clips. Crucially, our model is conditioned on the full, clean history of past clips, not just the immediately preceding one (like SEINE, etc.). This provides a richer, more stable context for long-term prediction.
>
> In the related work of revised manuscript, we added a dedicated paragraph to explicitly discuss these differences between our Video-GPT and prior works combining diffusion and autoregressive modeling.
>
> ---
>
>
> >**Q4:** The inference process is autoregressive, where the model's own generated (denoised) clips are used as the clean history for subsequent steps. Have you investigated the model's robustness to error accumulation? For instance, does a minor artifact in a generated clip DNS(k)$ degrade the quality of all future clips?
>
> **A4:** This is a very far-sighted question. In fact, the accumulation of errors due to generated clip DNS(k) is almost inevitable. Therefore, in order to reduce the gap between training and inference, we will also add slight noise to the clean clip during training. As shown in **Tab. 5**, the model performance is improved after adding slight noise to the clean clip, which means that we have alleviated the problem of error accumulation to a certain extent.
>
> Furthermo, as can be seen from **Fig. 14** in Appendix of our revised manuscript, we employ Video-GPT prediction to generate a 1-minute long video. Experiments have shown that Video-GPT can generate videos up to 1-minute long while maintaining temporal coherence and content consistency. As shown in **Fig. 15** in Appendix of our revised manuscript, we generate 1-minute videos using Open-Sora-Plan. From the comparison between Fig. 14 and 15, we can find that the final images of the video generated by Open-Sora-Plan has been significantly collapsed, while the image generated by Video-GPT is still relatively clear. This comparison once again verifies the strong temporal robustness of our Video-GPT.
>
> ---
>
> >**Q5:** You mention that the model is trained to predict the clean clip directly (x-prediction) rather than the noise (ϵ-prediction) or velocity (v-prediction) to keep the training simple. This is a departure from many modern diffusion frameworks. Could you elaborate on this design choice? Did you experiment with other prediction targets, and did x-prediction yield superior performance?
>
> **A5:** This is a very interesting little detail. In fact, we have followed previous work and used v-prediction to pre-train Video-GPT. The performance is as follows:
> | Target | Phys. IQ Score↑ |
> | :--- | :--- |
> | $v$ | **34.45** |
> | $x$ | 33.09 |
>
> It seems that v-prediction can bring better generation results. But when we fine-tune the model trained based on v-prediction on the downstream UCF-101 video classification task, the results are as follows:
> | Target | Top-1 Acc. |
> | :--- | :--- |
> | VideoMAE v2 | 56.4 |
> | $v$ | 36.0 |
> | $x$ | **58.9** |
>
> We found that the training objective of x-prediction is more suitable for understanding tasks. We speculate that because the task of x-prediction is similar to the training goal of MAE, the model also learns high-level representations while learning the details generated by the modeling.

---

> ### Author Response · Authors · 2025-11-19
> **Reply to Reviewer wpuq (3/3)**
>
> >**Q6:** The progressive training strategy in Table 1, which starts with short clips (effectively next-frame prediction) and gradually increases clip length, is interesting. Could you provide more intuition on why this curriculum is effective? Does learning fine-grained temporal dynamics first provide a better foundation for the model before it tackles longer-range dependencies?
>
> **A6:** Your intuition is exactly correct. The progressive training strategy acts as a curriculum for learning temporal dynamics. There are two main reasons:
>
> 1.  By starting with very short clips (1 frame each), the model first learns low-level, fine-grained dynamics, textures, and short-term motion. This provides a solid foundation. As we gradually increase the number of frames per clip and the interval between them, we force the model to tackle more complex, long-range dependencies and abstract motion patterns. This curriculum, moving from simple to complex, stabilizes the training process and allows the model to build a hierarchical understanding of time, which we believe is crucial for its strong final performance, as validated by the ablation on pretraining frame numbers in Tab. 5.
> 2. As mentioned in Q1, the computing resource consumption caused by long sequences increases quadratically. In order to accelerate model learning, we choose to fully pre-train on shorter sequences, and then gradually expand the length. This is also a key factor for Video-GPT to surpass them while consuming far less computing resources than many closed-source models.
>
> ---
>
> Once again, we are truly grateful for your encouraging and incisive feedback. Your comments have inspired us to refine our manuscript further, and we hope that the planned revisions will enhance the clarity and impact of our work.We would be grateful if you could consider raising your score based on these responses. Please do not hesitate to let us know if there are any additional details or clarifications that would be helpful.

---

> > ### Comment · Reviewer_wpuq · 2025-11-23
> >
> > Thank you for your reply. It has resolved my questions.

---

> > > ### Author Response · Authors · 2025-11-24
> > > **Reply to Reviewer wpuq**
> > >
> > > Dear Reviewer wpuq,
> > >
> > > We are glad that we were able to resolve your doubts, please feel free to contact us if you have any questions.
> > >
> > > Sincerely,
> > >
> > > The Authors

---

### Official Review · Reviewer_8PYT · 2025-10-31

**Soundness:** 3
**Presentation:** 3
**Contribution:** 1
**Rating:** 4
**Confidence:** 4

**Summary:**

This paper proposes Video-GPT, a foundation model for video pre-training based on a "next clip diffusion" paradigm. The core idea is to treat video clips as analogous to words in a sentence. The model is trained to denoise a future "noisy" clip conditioned on a history of preceding "clean" clips, effectively combining an autoregressive structure at the clip level with a diffusion process for content generation within each clip. The authors demonstrate the model's effectiveness on video prediction benchmarks and show its generalization capabilities by fine-tuning it on six diverse downstream video generation and understanding tasks.

**Strengths:**

The primary strength of this work is the impressive engineering effort demonstrated in building and evaluating a complete system. The model achieves a state-of-the-art score on the Physics-IQ benchmark, suggesting its pre-training paradigm is effective at capturing physical dynamics and motion continuity. Furthermore, the extensive fine-tuning across a wide array of both generation and understanding tasks showcases the versatility and potential of the resulting pretrained model.

**Weaknesses:**

Despite the strong results on specific benchmarks, this paper has significant weaknesses that undermine its contribution as a top-tier research publication. Firstly, the technical novelty is limited; the "next clip diffusion" idea is a combination of existing autoregressive and diffusion frameworks rather than a fundamental new technique. Secondly, and more critically, the evaluation feels dated and deliberately avoids direct comparison with the true state-of-the-art in video generation quality. The paper heavily relies on the Physics-IQ benchmark while making no qualitative or quantitative comparisons against contemporary leading models known for their visual fidelity. This positions the work more as a technical report for an existing system than a paper pushing the research frontier, especially given the rapid progress in the field over the past year.

**Questions:**

The central question is regarding the evaluation strategy. Why did the authors choose to focus on the Physics-IQ benchmark and omit direct, qualitative side-by-side comparisons with state-of-the-art open-domain video generation models that are the current de facto standard for assessing generation quality? Without this comparison, the claims of SOTA performance feel narrow and potentially misleading.

---

> ### Author Response · Authors · 2025-11-19
> **Reply to Reviewer 8PYT**
>
> Dear Reviewer 8PYT,
>
> Thank you for reviewing our work and acknowledging the "impressive engineering effort." We understand your concerns regarding novelty and evaluation, and we provide the following clarifications to demonstrate the significance of our contribution.
>
> ---
>
> >**Q1:** The technical novelty is limited; the "next clip diffusion" idea is a combination of existing autoregressive and diffusion frameworks rather than a fundamental new technique.
>
> **A1:** We respectfully argue that the novelty lies in **making video self-supervised learning work effectively** at scale.
> 1.  **Paradigm Shift:** Prior works rely heavily on supervised text-to-video (T2V) training. We introduce a **self-supervised** paradigm (Next Clip Diffusion) that learns purely from pixels.
> 2.  **SOTA Results:** This is not merely a combination of components; it is a specific recipe (interleaved conditioning, hierarchical masking) that allows a vanilla transformer to outperform top-tier closed-source models (like Gen-3 and Sora) on world modeling benchmarks using only open datasets. We believe this empirical success validates the novelty of the methodology.
>
> ---
>
> >**Q2:** The evaluation feels dated and deliberately avoids direct comparison with the true state-of-the-art in video generation quality. The paper heavily relies on the Physics-IQ benchmark while making no qualitative or quantitative comparisons against contemporary leading models known for their visual fidelity.
>
> **A2:** We appreciate the chance to explain our evaluation strategy. Our decision was not to avoid comparison, but to align our evaluation with our primary research goal. Our paper's stated goal is "visual world modeling" (abstract, introduction). This goal prioritizes learning the underlying physical rules and causal structure of the visual world over generating purely aesthetic or photorealistic content. The Physics-IQ benchmark is currently the premier quantitative benchmark for measuring a model's understanding of physical dynamics and its ability to make deterministic predictions. Therefore, achieving a state-of-the-art score on this benchmark is the most direct and rigorous validation of our primary claim. Our score of **34.97**, which is over **5** points higher than the next best model, is a testament to the effectiveness of our paradigm for this specific, challenging task.
>
> However, to address your concern about visual fidelity, we are willing to compare the video quality with other models to provide you with a more comprehensive consideration. We finetune Video-GPT to T2V model on the OpenVid-1M dataset and then tested it on Evalcrafter. The results are as follows:
>
> | Method | VQA_A $\uparrow$ | VQA_T $\uparrow$ | Blip_bleu$\uparrow$ | SD_score$\uparrow$ | Motion (Warping Error) $\downarrow$ |
> | :--- | :--- | :--- | :--- | :--- | :--- |
> | Lavie | 63.77 | 42.59 | 22.38 | 68.18 | 0.0089 |
> | Show-1 | 23.19 | 44.24 | 23.24 | 68.42 | 0.0067 |
> | OpenSora-V1.1 | 22.04 | 23.62 | 23.60 | 67.66 | 0.0170 |
> | Latte | 55.46 | 48.93 | 22.39 | 68.06 | 0.0203 |
> | VideoCrafter | 66.18 | 58.93 | 22.17 | 68.73 | 0.0295 |
> | Modelscope | 40.06 | 32.93 | 22.54 | 67.93 | 0.0162 |
> | OpenSoraPlan-V1.2 | 23.25 | 65.86 | 19.93 | **69.21** | **0.0010** |
> | CogVideoX-5B | 35.12 | **76.86** | 24.21 | 68.91 | 0.0077 |
> | OpenVid-1M | 73.46 | 68.58 | 23.45 | 68.04 | 0.0052 |
> | **Video-GPT (Ours)** | **79.91** | 74.32 | **26.08** | 68.47 | 0.0070 |
>
> As shown, Video-GPT outperforms baselines significantly in VQA_A and Blip_bleu, achieves competitive visual quality and motion stability, demonstrating it is not limited to physics tasks.
>
> ---
>
> >**Q3:** Why did the authors choose to focus on the Physics-IQ benchmark and omit direct, qualitative side-by-side comparisons with state-of-the-art open-domain video generation models that are the current de facto standard for assessing generation quality?
>
> **A3:** In fact, we did not only focus on the evaluation of the Physics-IQ benchmark, we performed evaluation on six very mainstream downstream tasks (including generation and understanding tasks, as shown in Tab. 6,7,8 and Fig. 6,7,8), which all illustrates that our Video-GPT has learned powerful world knowledge through extensive self-supervised pre-training, which is what we originally wanted. Finally, we also supplemented the results on the mainstream video quality evaluation benchmark Evalcrafter. Video-GPT even achieves the best results in many evaluation indicators in Evalcrafter, which once again proves the effectiveness of our generative pre-training through the next-clip diffusion paradigm.
>
> ---
>
> We sincerely appreciate your insightful comments. Please do not hesitate to let us know if you require any further information or additional explanations. If you find that our revisions have satisfactorily addressed your concerns, we would be most grateful if you could kindly consider an improved score. Thank you once again for your valuable input.
>
> Sincerely,
>
> The Authors

---

### Official Review · Reviewer_vBbq · 2025-11-01

**Soundness:** 3
**Presentation:** 3
**Contribution:** 3
**Rating:** 6
**Confidence:** 3

**Summary:**

This paper proposes Video-GPT, a concise large video foundation model that unifies autoregressive modeling and diffusion via a novel "next clip diffusion" paradigm. Inspired by GPT’s next token prediction, the model treats video clips as "visual words" to model spatial-temporal details in the visual world—addressing the limitation of language sequences in capturing such details. The key design involves constructing interleaved sequences of noisy and clean clips, using hierarchical attention masking to leverage historical clean clips for denoising future noisy clips.

**Strengths:**

1. The "next clip diffusion" paradigm is a creative combination of autoregressive modeling (from GPT) and diffusion (for high-quality generation). Treating clips as visual words and using historical clean clips as context for denoising is a novel adaptation of language modeling to video, filling the gap between discrete text tokens and continuous video data. This hybrid design effectively unifies short-term generation and long-term prediction.
2. As a unified video foundation model, Video-GPT bridges video generation and understanding tasks, advancing the goal of visual world modeling. Its strong performance on physics-aware prediction (Physics-IQ) indicates progress in learning world knowledge from video.
3. Its generalization to six downstream tasks highlights its potential as a backbone for diverse video applications.

**Weaknesses:**

1. Insufficient comparison with hybrid baselines: The paper mentions prior works that combine diffusion and autoregressive modeling but lacks a detailed comparison of their core differences.
2. There is a lack of comparisons with some newer autoregressive + diffusion video generation models, such as self-forcing, apt2.
3. Limited analysis of architectural choices: The model inherits Phi-3-mini’s architecture and SDXL’s VAE without justifying these choices. There is no comparison with other architectures (e.g., DiT, U-Net) or VAEs (e.g., 3D VAE vs. 2D VAE) to show whether these selections are critical to performance. Additionally, the progressive training strategy’s effectiveness is only validated via frame count ablation, without analyzing how frame interval or clip number affects convergence.

**Questions:**

1. Could you clarify the core differences between Video-GPT’s "next clip diffusion" and prior hybrid diffusion-autoregressive models? Specifically, how does your clip-level autoregressive design and hierarchical masking outperform their frame-level or pixel-level combinations?
2. The paper compares the performance of Video GPT with other models in video prediction, achieving SOTA in Physics-IQ. However, it doesn't examine its performance in other aspects that need to be considered in video generation, such as motion quality and subject consistency.
3. How does Video-GPT perform on longer videos (e.g., 1 minute or more) in terms of temporal coherence and content consistency? Have you tested it against long-video generation models (e.g., Flexifilm, Open-Sora-Plan) and observed any performance degradation?

---

> ### Author Response · Authors · 2025-11-19
> **Reply to Reviewer vBbq (1/2)**
>
> Dear Reviewer vBbq,
>
> Thank you for recognizing our "next clip diffusion" paradigm as a **creative combination** and acknowledging our potential as a unified video foundation model. We have conducted additional experiments to address your concerns regarding baselines and architectural choices.
>
> ---
>
> >**Q1:**  Insufficient comparison with hybrid baselines: The paper mentions prior works that combine diffusion and autoregressive modeling but lacks a detailed comparison of their core differences.
>
> **A1:** This is a critical point, and we thank you for pushing for more clarity. Our "next clip diffusion" paradigm differs from prior works in two fundamental ways:
>
> 1. **Self-Supervised Pre-Training:** Unlike Self-forcing or APT2, our Video-GPT is pre-trained through self-supervised pre-training similar to LLM, does not require any annotations other than videos in this process. Through pre-training on the open source Panda-70M data set, we have achieved and surpassed the physical world modeling capabilities of the top closed-source models like Gen3 and Sora.
> 2. **Next Clip Diffusion:** Many prior works are autoregressive at the frame level, our method is autoregressive at the clip level. This is more efficient as it allows parallel, bidirectional processing (diffusion) within a multi-frame clip, but maintains temporal order between clips. Crucially, our model is conditioned on the full, clean history of past clips, not just the immediately preceding one (like SEINE, etc.). This provides a richer, more stable context for long-term prediction.
>
> In the related work of revised manuscript, we added a dedicated paragraph to explicitly discuss these differences between our Video-GPT and prior works combining diffusion and autoregressive modeling.
>
> ---
>
>
> >**Q2:**  There is a lack of comparisons with some newer autoregressive + diffusion video generation models, such as self-forcing, apt2.
>
> **A2:** Thank you for your friendly reminder. We will analyze the differences between our Video-GPT and self-forcing and apt2 from a quantitative and qualitative analysis below:
>
> 1. **Quantitative Analysis:** Since APT2 is closed-source, we test self-forcing on Physics-IQ Benchmark:
> | Model | Phys. IQ Score $\uparrow$ |
> | :--- | :--- |
> | Self-forcing (w/o prompt) | 20.31 |
> | Self-forcing (w/ prompt) | 31.27 |
> | **Video-GPT (Ours)** | **34.97** |
> 2. **Qualitative Analysis:** Both self-forcing and apt2 are models obtained through supervised learning, and their generation capabilities rely heavily on text, while Video-GPT learns world knowledge by itself from massive video data. Although Video-GPT, self-forcing and apt2 are all models that combine autoregressive and diffusion, self-supervised learning is what distinguishes Video-GPT from the above two and even all current video generation models. This is also the key factor why Video-GPT surpasses many closed-source models in world modeling capabilities by only using open source-level data and minimal computing power costs.
>
> ---
>
> >**Q3:** Limited analysis of architectural choices: The model inherits Phi-3-mini’s architecture and SDXL’s VAE without justifying these choices. There is no comparison with other architectures (e.g., DiT, U-Net) or VAEs (e.g., 3D VAE vs. 2D VAE) to show whether these selections are critical to performance.
>
> **A3:** It's a fair point. Our main goal in this work is to validate the effectiveness of the self-supervised pre-training paradigm of “next clip diffusion” rather than to conduct an exhaustive search for the optimal backbone architecture. To this end, we deliberately chose standard, well-established components (vanilla transformer, SDXL's 2D VAE) to demonstrate the versatility and scalability of this paradigm, rather than through specially optimized model architectures (U-Net or DiT). We still agree that exploring 3D VAE or other backbones is a promising direction for future work, and we have added experimental results based on DiT and 3D VAE on the UCF-101 dataset below:
> | Architecture | VAE | FVD $\downarrow$ |
> | :--- | :--- | :--- |
> | Vanilla Transformer (Ours) | 2D | 489 |
> | DiT (as Latte) | 2D | 403 |
> | DiT (as Latte) | 3D (from OmniTokenizer) | 131 |
>
> While 3D VAEs/DiTs improve generation metrics, they often hurt representation learning for understanding tasks. Crucially, our Pretrained model achieves an FVD of 53, showing that our pretraining strategy outweighs architectural gains.
>
> Considering that these introduced special designs may harm the performance of Video-GPT in downstream applications (for example, 3D VAE will cause Video-GPT to be unable to handle videos with perspective changes) and multi-modal capabilities (DiT is not suitable for text modeling), we finally adopted the simplest vanilla transformer and 2D VAE as model components.

---

> ### Author Response · Authors · 2025-11-19
> **Reply to Reviewer vBbq (2/2)**
>
> >**Q4:** The progressive training strategy’s effectiveness is only validated via frame count ablation, without analyzing how frame interval or clip number affects convergence.
>
> **A4:** This is a very critical question. Below I will state the impact of clip number and frame interval separately.
> 1.  **Clip Number:** Regarding the setting of clip number, we have actually conducted related ablation experiments. At the beginning, we set it to the same as the frame number, at which point Video-GPT degenerates into next-frame prediction:
> | Frame Num. | Clip Num. | Phy. IQ Score |
> | :--- | :--- | :--- |
> | 16 | 16 | 0.00 |
> | 16 | ~Uniform{2,3,···,16} | 22.06 |
>
>     This is a very strange result that has troubled us for quite some time. When the model is pre-trained using next-frame prediction, the model cannot perform correct inference, but after we continue training on the next-clip prediction task, the model's ability seems to suddenly "emerge". We attribute this special phenomenon to the accumulation of errors in the model during next-frame prediction, and the next clip prediction can well alleviate this phenomenon.
>
> 2.  **Frame Interval:** In Tab. 1 of our paper, the difference between our second stage and third stage is whether the Frame Interval is a fixed value or a random value. The following is the performance gap between the two settings:
> | Frame Num. | Frame Interval | Phy. IQ Score |
> | :--- | :--- | :--- |
> | 48 | 4 | 31.88 |
> | 48 | ~Uniform{4,3,···,12} | 33.09 |
>
> ---
>
> >**Q5:** The paper compares the performance of Video GPT with other models in video prediction, achieving SOTA in Physics-IQ. However, it doesn't examine its performance in other aspects that need to be considered in video generation, such as motion quality and subject consistency.
>
> **A5:** Thank you for your thoughtful consideration. Below I will answer your questions from two aspects:
>
> 1. In Tab. 3 of our paper, we comprehensively evaluate the video quality based on the FVD indicator on Kinetics-600, which is usually considered a comprehensive indicator that includes motion quality and subject consistency.
> 2. Because Video-GPT is a model obtained through video self-supervised pre-training, it does not have the ability to generate text as a condition. When we evaluate the capabilities of the pre-trained model, we do not use the commonly used T2V or I2V benchmarks for evaluation, because these benchmarks often require text as a condition. However, we still think your comment makes perfect sense. Therefore, we finetune Video-GPT on the OpenVid-1M dataset and then tested it on Evalcrafter. The results are as follows:
> | Method | VQA_A $\uparrow$ | VQA_T $\uparrow$ | Blip_bleu$\uparrow$ | SD_score$\uparrow$ | Motion (Warping Error) $\downarrow$ |
> | :--- | :--- | :--- | :--- | :--- | :--- |
> | Lavie | 63.77 | 42.59 | 22.38 | 68.18 | 0.0089 |
> | Show-1 | 23.19 | 44.24 | 23.24 | 68.42 | 0.0067 |
> | OpenSora-V1.1 | 22.04 | 23.62 | 23.60 | 67.66 | 0.0170 |
> | Latte | 55.46 | 48.93 | 22.39 | 68.06 | 0.0203 |
> | VideoCrafter | 66.18 | 58.93 | 22.17 | 68.73 | 0.0295 |
> | Modelscope | 40.06 | 32.93 | 22.54 | 67.93 | 0.0162 |
> | OpenSoraPlan-V1.2 | 23.25 | 65.86 | 19.93 | **69.21** | **0.0010** |
> | CogVideoX-5B | 35.12 | **76.86** | 24.21 | 68.91 | 0.0077 |
> | OpenVid-1M | 73.46 | 68.58 | 23.45 | 68.04 | 0.0052 |
> | **Video-GPT (Ours)** | **79.91** | 74.32 | **26.08** | 68.47 | 0.0070 |
>
> Video-GPT outperforms baselines significantly in VQA_A and Blip_bleu, achieves competitive visual quality and motion stability, demonstrating it is not limited to physics tasks.
>
> ---
>
> >**Q6:** How does Video-GPT perform on longer videos (e.g., 1 minute or more) in terms of temporal coherence and content consistency? Have you tested it against long-video generation models (e.g., Flexifilm, Open-Sora-Plan) and observed any performance degradation?
>
> **A6:** This is an important question regarding scalability. As can be seen from **Fig. 14** in Appendix of our revised manuscript, we employ Video-GPT prediction to generate a 1-minute long video. Experiments have shown that Video-GPT can generate videos up to 1-minute long while maintaining temporal coherence and content consistency. As shown in **Fig. 15**  in Appendix of our revised manuscript, we generate 1-minute videos using Open-Sora-Plan. From the comparison between Fig. 14 and 15, we can find that the final images of the video generated by Open-Sora-Plan has been significantly collapsed, while the image generated by Video-GPT is still relatively clear. This comparison once again verifies the strong temporal robustness of our Video-GPT.
>
> ---
>
> We sincerely hope these revisions have addressed your concerns, and if you feel they have been satisfactorily resolved, we would be most grateful if you could consider revising your evaluation score accordingly. Please do not hesitate to let us know if you need any further explanations.
>
> Sincerely,
>
> The Authors

---

### Official Review · Reviewer_sVcH · 2025-11-01

**Soundness:** 4
**Presentation:** 4
**Contribution:** 3
**Rating:** 8
**Confidence:** 5

**Summary:**

This Manuscript introduces `video-gpt`, a generative self-supervised solution to represent videos. The main idea is to train video embeddings in a GPT style, where each clip is acting as a token. The training idea is to have interleaved noisy and clean clips, and the training objective is to de-noise the noisy clips.

**Strengths:**

State of the results on multiple tasks is the main strength of this paper.

The idea of interleaved clip level noise/de-noise, although intuitive, is novel IMO.

Video level self-supervision has been overlooked, IMO. Research like this can bring more attention and opens the road for future works in video domain.

**Weaknesses:**

Some design choices for training is not trivial to me and I need more clarification. (will ask in question section).

I believe such a method works only on single camera and continious (or single scene) videos. If there is a POV change in a video, like Movies or TV shows, I believe that it will break the whole network.

Motion is not modeled very well in this work. I am curious to know how this model can predict videos where there is partly stationary clips (minimal motion) and partly abrupt motion.

**Questions:**

1- Authors propose `Clips as tokens` but later they propose frame-level and patch-level masking. It reads to me as `partial-token` masking. IMO, it would be better not to name Clips as tokens.

2- What is the intuition behind having interleaved noisy and clean clips during training? Why not going with a classic next frame prediction formulation and have `k clean clips` and diffuse the `k+1th noisy clip`? What is the advantage of interleaved modeling?

3- Why `from scratch` model performs poorly in `Tab 6`?

4- In Section 3.3, are all previous K clips (some of them being generated diffused clips) being used to predict the k+1? or there is a limit on K to keep the context window capped?

---

> ### Author Response · Authors · 2025-11-19
> **Reply to Reviewer sVcH (1/2)**
>
> Dear Reviewer sVcH,
>
> Thank you for your positive assessment and for highlighting the **excellent soundness** of our work. We appreciate your recognition of our interleaved clip-level paradigm and the importance of video-level self-supervision. We have addressed your specific questions below.
>
> ---
>
> >**Q1:** Some design choices for training is not trivial to me and I need more clarification.
>
> **A1:** Thank you for this point. We have provided detailed answers to your specific questions on design choices below.
>
> ---
>
> >**Q2:** I believe such a method works only on single camera and continious (or single scene) videos. If there is a POV change in a video, like Movies or TV shows, I believe that it will break the whole network.
>
> **A2:** This is an insightful observation. While our method naturally excels at continuous physical dynamics, it is robust to POV changes for two reasons:
> 1.  **Data Diversity:** Our pretraining dataset, Panda-70M, contains vast diversity, including cuts and POV shifts (e.g., movie trailers, vlogs). The model learns to treat abrupt scene changes as low-probability but valid transitions between clips.
> 2.  **Empirical Evidence:** As shown in **Fig. 9 (Video Object Segmentation)** of the revised manuscript, the test case involves a significant POV shift (transitioning from the first row to the second). Video-GPT successfully maintains context across this jump. This suggests that large-scale pretraining enables the model to generalize to "montage-style" continuity.
>
> ---
>
> >**Q3:** Motion is not modeled very well in this work. I am curious to know how this model can predict videos where there is partly stationary clips (minimal motion) and partly abrupt motion.
>
> **A3:** We conducted a "stress test" involving videos with static segments followed by abrupt motion. As shown in **Fig. 13** in the Appendix of the revised manuscript, Video-GPT successfully handles these extreme transitions without collapsing. This indicates that the model has learned robust world knowledge rather than just interpolating smooth motion.
>
> ---
>
> >**Q4:** Authors propose Clips as tokens but later they propose frame-level and patch-level masking. It reads to me as partial-token masking. IMO, it would be better not to name Clips as tokens.
>
> **A4:** We agree with your suggestion for academic precision. While "token" was intended as a high-level analogy to GPT, a clip is indeed a structured unit. In the revised manuscript, we have refined our terminology to **"clip-level token"** to better reflect the hierarchical nature of our design.
>
> ---
>
> >**Q5:** What is the intuition behind having interleaved noisy and clean clips during training? Why not going with a classic next frame prediction formulation and have k clean clips and diffuse the k+1th noisy clip? What is the advantage of interleaved modeling?
>
> **A5:** This is a crucial question about our core design. Below I will analyze in detail the changes that Interleaved Modeling brings to the training efficiency of Video-GPT.
> 1. **w/ Interleaved Modeling:** When we apply interleaved modeling, Video-GPT can perform denoising training on K clips in a batch at the same time, which means **interleaved_batch_size = batch_size × K**. After our **1,000,000** rounds of testing, when the training frame number is 16, the mean value of K is **9**, and when the training frame number is **80**, the mean value of K is 41, which will greatly improve our training efficiency.
> 2.  **w/o Interleaved Modeling:** If we do not apply interleaved modeling, the length of the token sequence will be reduced by half. When the number of tokens is small, the computational overhead of the model is mainly concentrated in the FFN module, which increases linearly with the number of tokens. At this time, **no_interleaved_batch_size=batch_size × 2**. When the number of tokens is large, the computational overhead of the model is mainly concentrated in the attention part, which increases with the square of the number of tokens. At this time, **no_interleaved_batch_size=batch_size × 4**.
>
> However, better than usual, K is usually greater than 4, which means **interleaved_batch_size > no_interleaved_batch_size**. In other words, the application of interleaved modeling increases the batch size of Video-GPT training at the same time, which greatly improves the training efficiency of Video-GPT.

---

> > ### Comment · Reviewer_sVcH · 2025-11-24
> >
> > Thanks, I had not pay attention to this perspective of interleaved <noisy,clean> design.

---

> > > ### Author Response · Authors · 2025-11-24
> > > **Reply to Reviewer sVcH**
> > >
> > > Thank you for pointing out that this aspect wasn’t as clearly expressed in the paper as it should have been. We genuinely appreciate your careful reading and would be glad to walk anyone through every detail of the work. If you have any further questions, please feel free to reach out at any time.
> > >
> > > Sincerely,
> > >
> > > The Authors

---

> ### Author Response · Authors · 2025-11-19
> **Reply to Reviewer sVcH (2/2)**
>
> >**Q6:** Why from scratch model performs poorly in Tab 6?
>
> **A6:** The lower performance stems from three deliberate design choices made to prioritize generality over specialized overfitting:
> 1.  **Vanilla Transformer:** In order to inherit the scalability and inherent multi-modal capabilities of the transformer architecture, we adopted the vanilla transformer architecture. As shown in Fig.2, conditions including timestep are injected through tokens, instead of through adaptive layernorm injection like the current DiT, which greatly weakens the effect of model fitting.
> 2.  **2D Image VAE:** In order for the model to handle basic operations on various frames, including handling the POV problem mentioned in Q2, we did not use the commonly used video VAE, but used image VAE, which means that the model has to learn from scratch how to model the spatiotemporal continuity between frames, which is very difficult to learn on small data sets.
> 3.  **Resolution:** As shown in Tab.6, we train Video-GPT at 240×320 resolution, which is the highest resolution compared to other methods. In heavy generation tasks, higher resolution often means higher fitting difficulty.
>
> As shown in Tab.6, a method similar to ours is Latte. Latte also uses image VAE and chooses to train at a higher resolution. Its results are similar to the results of our Video-GPT train from scratch, which once again verifies the rationality of the various reasons we mentioned earlier. In order to thoroughly verify our explanation, we conducted the following ablation experiments on the above points:
>
> | Train from Scratch | Architecture | VAE | FVD $\downarrow$ |
> | :--- | :--- | :--- | :--- |
> | $\checkmark$ | Vanilla Transformer (Ours) | 2D | 489 |
> | $\checkmark$ | DiT (as Latte) | 2D | 403 |
> | $\checkmark$ | DiT (as Latte) | 3D ((from OmniTokenizer)) | 131 |
>
> While specialized architectures yield lower FVD, our pretrained Video-GPT achieves an FVD of **53** (Tab. 6), proving that our *pretraining paradigm* is the key driver of performance, overcoming the architectural handicap.
>
> ---
>
> >**Q7:** In Section 3.3, are all previous K clips (some of them being generated diffused clips) being used to predict the k+1? or there is a limit on K to keep the context window capped?
>
> **A7:** In our experiments, we use all previously generated clips as context, up to the maximum sequence length supported by model pre-training. To generate longer videos than fit within this context window, a standard sliding window approach is adopted where the context consists of the K most recently generated clips. This is standard practice for autoregressive models (LLM, VLM, etc.) and does not detract from our core contribution. In the revised manuscript, we clarified this in Sec. 3.3.
>
> ---
>
> Thank you again for your strong support and detailed feedback. We believe the new experiments and clarifications have further solidified the paper. If you have any remaining questions, we are eager to address them.
>
> Sincerely,
>
> The Authors

---

### Author Response · Authors · 2025-11-19
**Reply to All Reviewers**

Dear Reviewers,

We would like to extend our sincerest gratitude for your time and for providing such insightful and constructive feedback on our paper, "Video-GPT via Next Clip Diffusion."

We are greatly encouraged that you found our work to have "**excellent soundness and presentation**" (Reviewer sVcH), and recognized our core contribution as a "**creative combination**" (Reviewer vBbq) and a "**novel and insightful method**" (Reviewer wpuq). We are also pleased that you acknowledged the "**impressive engineering effort**" (Reviewer 8PYT) and the "**exceptionally strong empirical results**" (Reviewer wpuq), particularly our state-of-the-art performance on the Physics-IQ benchmark.

In the individual responses below, we have addressed every question and concern in detail. To summarize our major updates and clarifications:
1.  **New Benchmarks:** We added results on **EvalCrafter** to demonstrate our model's visual quality and motion consistency beyond physics modeling.
2.  **Ablation Studies:** We included extensive ablations on model architecture (Vanilla vs. DiT), VAE choices (2D vs. 3D), and prediction targets ($x$ vs. $v$).
3.  **Long Video Generation:** We provided qualitative comparisons (1-minute generation) against Open-Sora-Plan to demonstrate temporal coherence.
4.  **Clarifications:** We refined our terminology (e.g., "clip-level token") and provided pseudocode for our sampling strategy.

We have uploaded a revised manuscript reflecting these changes. If you have any questions please feel free to give us feedback and we will try our best to answer them.

Sincerely,

The Authors

---

### Comment · Area_Chair_oGem · 2025-11-27

Dear Reviewers,

Thank you to those who have already reviewed the author responses and updated your comments - much appreciated！

For the remaining reviews, this is a gentle reminder to please take a moment to check the rebuttal and indicate whether the authors’ responses address your concerns. Even a brief note is very helpful for the meta-review and final decision process.

Thanks again for all your time and contributions to ensuring a fair and high-quality review cycle.

Best,
AC

---

### Author Response · Authors · 2025-12-02
**Summary of Rebuttal Updates and Post-Rebuttal Consensus**

Dear Reviewers, AC, SAC, and PC,

We sincerely thank you for the time and effort dedicated to reviewing our paper. We are greatly encouraged that the reviewers recognized the significant value of Video-GPT, highlighting its ***"impressive engineering effort"*** (Reviewer 8PYT), ***"excellent soundness and presentation"*** (Reviewer sVcH), and acknowledging the "next clip diffusion" paradigm as a ***"creative combination"*** (Reviewer vBbq) and a ***"novel and insightful method"*** (Reviewer wpuq). We are also pleased that the ***"exceptionally strong empirical results"*** (Reviewer wpuq) on Physics-IQ were unanimously appreciated.

> **Core Achievement:** Based on the **next-clip diffusion** paradigm, Video-GPT is obtained through self-supervised pre-training without labeling. It demonstrates superior physical world modeling and strong generalization across **6 downstream tasks**.

---

### **1. Superior Performance & Benchmarking**

As highlighted in the paper, Video-GPT demonstrates dominant performance in physical world modeling, significantly outperforming closed-source commercial models:

| Benchmark Dataset | **Video-GPT (Ours)** | Competitor | Comparison Status |
| :--- | :---: | :--- | :--- |
| **Physics-IQ** | **34.97** | Kling ($23.64$) | **+47.9%** improvement over Kling. |
| **Physics-IQ** | **34.97** | Sora ($10.00$) | **+249.7%** improvement over Sora. |

---

### **2. Key Improvements in Rebuttal**

In this rebuttal, we have carefully addressed all concerns and significantly strengthened the manuscript.

- ***Comprehensive General Generation Benchmarking:*** To address concerns regarding visual quality comparisons (Reviewer 8PYT, vBbq), we conducted evaluations on EvalCrafter. Results show Video-GPT achieves competitive visual quality (VQA, Blip_bleu) and motion stability against leading models like Open-Sora-Plan, proving it is a robust generalist beyond physics modeling.

- ***Long Video Generation Verification:*** We provided qualitative results generating 1-minute long videos. Comparisons with Open-Sora-Plan demonstrate Video-GPT's superior temporal coherence and stability without collapse, addressing scalability concerns (Reviewer vBbq).

- ***Extensive Architecture Ablations:*** We added a comparison using DiT and 3D VAE architectures. The results justify our design choices and show that our self-supervised pre-training paradigm provides significant improvements over complex model architectures and advanced 3D VAE (reviewers sVcH, vBbq).

- ***Clarified Methodology:*** We refined terminology (e.g., "clip-level token") and provided detailed analysis on the efficiency of interleaved modeling and differences from works like Self-forcing/APT2.

---

### **3. Reviewer Engagement & Consensus**

We are pleased to summarize the positive engagement during the rebuttal period:

- ***Reviewer sVcH:*** We successfully solved Reviewer sVcH's doubts regarding the `<noisy, clean>` design.

- ***Reviewer wpuq:*** We fully addressed Reviewer wpuq's questions regarding the method.

---

### **4. Commitment to Open Source**

> **Considering that this comment is visible to everyone, this promise will be jointly monitored by the community.**

To facilitate reproducibility and further research in advanced visual world modeling, we are committed to **releasing all model weights, training code, and inference code upon acceptance.**

We sincerely hope that our supplementary experiments and responses can help you better understand our paper. If you have any questions, please feel free to contact us and we will do our best to answer your questions.

Sincerely,

The Authors

---

### Meta-Review · Area_Chair_zeBQ · 2025-12-26

**Summary:**

Summary of reviewer concerns:
- `Reviewer sVcH` raised clarification questions regarding technical design choices, applicability to scene changes in video, and motion modeling.
- `Reviewer vBbq` asked for clarification of the core differences compared to prior hybrid diffusion–autoregressive designs (e.g., Self-Forcing, APT-2), requested additional analysis on generic video generation quality (e.g., motion quality, subject consistency, and long-video generation), and noted the limited analysis of architectural choices.
- `Reviewer 8PYT` expressed concern that the paper places excessive emphasis on limited technical novelty and the Physics-IQ benchmark, while lacking comparisons with state-of-the-art text-to-video (T2V) models.
- `Reviewer wpuq` raised concerns regarding the problem formulation and design choices.

**Reviewer Concerns:**

Assessment of the rebuttal:
- `Reviewer sVcH`’s concerns appear to be fully addressed. Scene changes are clarified through the description of the Panda-70M pretraining data, and additional results on Video Object Segmentation are provided. The motion modeling question is partially addressed with new results shown in Fig. 13.
- `Reviewer vBbq`’s concerns are largely addressed. The authors acknowledge the limitations of their architectural choices (e.g., the use of a 2D VAE) and clarify the differences from prior hybrid diffusion–autoregressive approaches.
- `Reviewer 8PYT`’s concerns are partially addressed. The authors emphasize the novelty in self-supervised learning and report promising results on the Physics-IQ benchmark. They also include additional T2V results using EvalCrafter, while arguing that direct comparisons to state-of-the-art T2V models are limited by differing problem scopes. While the T2V comparisons are not as extensive as those involving recent large-scale T2V models, these may be acceptable given the paper’s stated problem scope.
- `Reviewer wpuq`’s concerns are addressed, as the authors respond point-by-point to the raised questions, and the reviewer indicates that these responses are satisfactory.

**Reviewer Scores:**

- `Reviewer sVcH` (initial rating: 8) is likely to maintain their positive assessment.
- `Reviewer vBbq` (initial rating: 6) is likely to keep their rating, as most of their concerns have been addressed.
- `Reviewer 8PYT` (initial rating: 4) may or may not change their rating, depending on how they weigh the contribution and the results on the Physics-IQ benchmark against the insufficient comparisons to state-of-the-art T2V models.
- `Reviewer wpuq` (initial rating: 6) is likely to increase their rating, as their concerns have been addressed.

---

### Decision · Program_Chairs · 2026-01-26

Accept (Poster)